# Learning to Utilize Shaping Rewards:
# A New Approach of Reward Shaping

**Yujing Hu**[1], **Weixun Wang**[1,2], **Hangtian Jia**[1], **Yixiang Wang**[1,3], **Yingfeng Chen**[1]*,
**Jianye Hao**[2,4], **Feng Wu**[3], **Changjie Fan**[1]

[1]Netease Fuxi AI Lab, Netease, Inc., Hangzhou, China
[2]College of Intelligence and Computing, Tianjin University, Tianjin, China
[3]School of Computer Science and Technology, University of Science and Technology of China
[4]Noah's Ark Lab, Huawei, China

huyujing@corp.netease.com, wxwang@tju.edu.cn, jiahangtian@corp.netease.com
yixiangw@mail.ustc.edu.cn, chenyingfeng1@corp.netease.com
jianye.hao@tju.edu.cn, wufeng02@ustc.edu.cn, fanchangjie@corp.netease.com

## Abstract

Reward shaping is an effective technique for incorporating domain knowledge into reinforcement learning (RL). Existing approaches such as potential-based reward shaping normally make full use of a given shaping reward function. However, since the transformation of human knowledge into numeric reward values is often imperfect due to reasons such as human cognitive bias, completely utilizing the shaping reward function may fail to improve the performance of RL algorithms. In this paper, we consider the problem of adaptively utilizing a given shaping reward function. We formulate the utilization of shaping rewards as a bi-level optimization problem, where the lower level is to optimize policy using the shaping rewards and the upper level is to optimize a parameterized shaping weight function for true reward maximization. We formally derive the gradient of the expected true reward with respect to the shaping weight function parameters and accordingly propose three learning algorithms based on different assumptions. Experiments in sparse-reward *cartpole* and *MuJoCo* environments show that our algorithms can fully exploit beneficial shaping rewards, and meanwhile ignore unbeneficial shaping rewards or even transform them into beneficial ones.

## 1 Introduction

A common way for addressing the sample efficiency issue of RL is to transform possible domain knowledge into additional rewards and guide learning algorithms to learn faster and better with the combination of the original and new rewards, which is known as reward shaping (RS). Early work of reward shaping can be dated back to the attempt of using hand-crafted reward function for robot behavior learning [3] and bicycle driving [15]. The most well-known work in the reward shaping domain is the potential-based reward shaping (PBRS) method [12], which is the first to show that policy invariance can be guaranteed if the shaping reward function is in the form of the difference of potential values. Recently, reward shaping has also been successfully applied in more complex problems such as Doom [8, 18] and Dota 2 [13].

Existing reward shaping approaches such as PBRS and its variants [2, 5, 6, 24] mainly focus on the way of generating shaping rewards (e.g., using potential values) and normally assume that the shaping rewards transformed from prior knowledge are completely helpful. However, such an assumption is not practical since the transformation of human knowledge (e.g., rules) into numeric

---

values (e.g., rewards or potentials) inevitably involves human operations, which are often subjective and may introduce human cognitive bias. For example, for training a Doom agent, designers should set appropriate rewards for key events such as object pickup, shooting, losing health, and losing ammo [8]. However, there are actually no instructions indicating what specific reward values are appropriate and the designers most probably have to try many versions of reward functions before getting a well-performed agent. Furthermore, the prior knowledge provided may also be unreliable if the reward designers are not experts of Doom.

Instead of studying how to generate useful shaping rewards, in this paper, we consider how to adaptively utilize a given shaping reward function. The term *adaptively utilize* stands for utilizing the beneficial part of the given shaping reward function as much as possible and meanwhile ignoring the unbeneficial shaping rewards. The main contributions of this paper are as follows. Firstly, we define the utilization of a shaping reward function as a bi-level optimization problem, where the lower level is to optimize policy for shaping rewards maximization and the upper level is to optimize a parameterized shaping weight function for maximizing the expected accumulative true reward. Secondly, we provide formal results for computing the gradient of the expected true reward with respect to the weight function parameters and propose three learning algorithms for solving the bi-level optimization problem. Lastly, extensive experiments are conducted in *cart-pole* and *MuJoCo* environments. The results show that our algorithms can identify the quality of given shaping rewards and adaptively make use of them. In some tests, our algorithms even transform harmful shaping rewards into beneficial ones and help to optimize the policy better and faster.

## 2 Background

In this paper, we consider the policy gradient framework [21] of reinforcement learning (RL) and adopt Markov decision process (MDP) as the mathematical model. Formally, an MDP is a tuple $\mathcal{M} = \langle \mathcal{S}, \mathcal{A}, P, r, p_0, \gamma \rangle$, where $\mathcal{S}$ is the state space, $\mathcal{A}$ is the action space, $P : \mathcal{S} \times \mathcal{A} \times \mathcal{S} \to [0, 1]$ is the state transition function, $r : \mathcal{S} \times \mathcal{A} \to \mathbb{R}$ is the (expected) reward function, $p_0 : \mathcal{S} \to [0, 1]$ is the probability distribution of initial states, and $\gamma \in [0, 1)$ is the discount rate. Generally, the policy of an agent in an MDP is a mapping $\pi : \mathcal{S} \times \mathcal{A} \to [0, 1]$ and can be represented by a parameterized function (e.g., a neural network). In this paper, we denote a policy by $\pi_\theta$, where $\theta$ is the parameter of the policy function. Let $p(s' \to s, t, \pi_\theta)$ be the probability that state $s$ is visited after $t$ steps from state $s'$ under the policy $\pi_\theta$. We define the discounted state distribution $\rho^\pi(s) = \int_{\mathcal{S}} \sum_{t=1}^{\infty} \gamma^{t-1} p_0(s') p(s' \to s, t, \pi_\theta) \mathrm{d}s'$, which is also called the discounted weighting of states when following. The goal of the agent is to optimize the parameter $\theta$ for maximizing the expected accumulative rewards $J(\pi_\theta) = \mathbb{E}_{s \sim \rho^\pi, a \sim \pi_\theta} [r(s, a)]$. According to the policy gradient theorem [21], the gradient of $J(\pi_\theta)$ with respect to $\theta$ is $\nabla_\theta J(\pi_\theta) = \mathbb{E}_{s \sim \rho^\pi, a \sim \pi_\theta} [\nabla_\theta \log \pi_\theta(s, a) Q^\pi(s, a)]$, where $Q^\pi$ is the state-action value function.

### 2.1 Reward Shaping

Reward shaping refers to modifying the original reward function with a shaping reward function which incorporates domain knowledge. We consider the most general form, namely the additive form, of reward shaping. Formally, this can be defined as $r' = r + F$, where $r$ is the original reward function, $F$ is the shaping reward function, and $r'$ is the modified reward function. Early work of reward shaping [3, 15] focuses on designing the shaping reward function $F$, but ignores that the shaping rewards may change the optimal policy. Potential-based reward shaping (PBRS) [12] is the first approach which guarantees the so-called policy invariance property. Specifically, PBRS defines $F$ as the difference of potential values: $F(s, a, s') = \gamma \Phi(s') - \Phi(s)$, where $\Phi : \mathcal{S} \to \mathbb{R}$ is a potential function which gives some kind of hints on states. Important variants of PBRS include the potential-based advice (PBA) approach: $F(s, a, s', a') = \gamma \Phi(s', a') - \Phi(s, a)$, which defines $\Phi$ over the state-action space for providing advice on actions [24], the dynamic PBRS approach: $F(s, t, s', t') = \gamma \Phi(s', t') - \Phi(s, t)$, which introduces a time parameter into $\Phi$ for allowing dynamic potentials [2], and the dynamic potential-based advice (DPBA) approach which learns an auxiliary value function for transforming any given rewards into potentials [6].

## 2.2 Related Work

Besides the shaping approaches mentioned above, other important works of reward shaping include the theoretical analysis of PBRS [23, 9], the automatic shaping approaches [11, 5], multi-agent reward shaping [1, 20], and some novel approaches such as belief reward shaping [10], ethics shaping [25], and reward shaping via meta learning [28]. Similar to our work, the automatic successive reinforcement learning (ASR) framework [4] learns to take advantage of multiple auxiliary shaping reward functions by optimizing the weight vector of the reward functions. However, ASR assumes that all shaping reward functions are helpful and requires the weight sum of these functions to be one. In contrast, our shaping approaches do not make such assumptions and the weights of the shaping rewards are state-wise (or state-action pair-wise) rather than function-wise. Our work is also similar to the optimal reward framework [17, 19, 27] which maximizes (extrinsic) reward by learning an intrinsic reward function and simultaneously optimizing policy using the learnt reward function. Recently, Zheng *et al.* [26] extend this framework to learn intrinsic reward function which can maximize lifetime returns and show that it is feasible to capture knowledge about long-term exploration and exploitation into a reward function. The most similar work to our paper may be the population-based method [7] which adopts a two-tier optimization process to learn the intrinsic reward signals of important game points and achieves human-level performance in the capture-the-flag game mode of Quake III. Learning an intrinsic reward function has proven to be an effective way for improving the performance of RL [14] and can be treated as a special type of online reward shaping. Instead of investigating how to learn helpful shaping rewards, our work studies a different problem where a shaping reward function is available, but how to utilize the function should be learnt.

## 3 Parameterized Reward Shaping

Given an MDP $\mathcal{M} = \langle \mathcal{S}, \mathcal{A}, P, r, p_0, \gamma \rangle$ and a shaping reward function $f$, our goal is to distinguish between the beneficial and unbeneficial rewards provided by $f$ and utilize them differently when optimizing policy in $\mathcal{M}$. By introducing a *shaping weight function* into the additive form of reward shaping, the utilization problem of shaping rewards can be modeled as

$$\tilde{r}(s,a) = r(s,a) + z_\phi(s,a)f(s,a), \tag{1}$$

where $z_\phi : \mathcal{S} \times \mathcal{A} \rightarrow \mathbb{R}$ is the shaping weight function which assigns a weight to each state-action pair and is parameterized by $\phi$. In our setting, the shaping reward function is represented by $f$ rather than $F$ for keeping consistency with the other lower-case notations of rewards. We call this new form of reward shaping *parameterized reward shaping* as $z_\phi$ is a parameterized function. For the problems with multiple shaping reward functions, $z_\phi(s,a)$ is a weight vector where each element corresponds to one shaping reward function.

### 3.1 Bi-level Optimization

Let $\pi_\theta$ denote an agent's (stochastic) policy with parameter $\theta$. The learning objective of parameterized reward shaping is twofold. Firstly, the policy $\pi_\theta$ should be optimized according to the modified reward function $\tilde{r}$. Given the shaping reward function $f$ and the shaping weight function $z_\phi$, this can be defined as $\tilde{J}(\pi_\theta) = \mathbb{E}_{s\sim\rho^\pi, a\sim\pi_\theta}\left[r(s,a) + z_\phi(s,a)f(s,a)\right]$. Secondly, the shaping weight function $z_\phi$ needs to be optimized so that the policy $\pi_\theta$, which maximizes $\tilde{J}(\pi_\theta)$, can also maximize the expected accumulative true reward $J(z_\phi) = \mathbb{E}_{s\sim\rho^\pi, a\sim\pi_\theta}\left[r(s,a)\right]$. The idea behind $J(z_\phi)$ is that although $z_\phi$ cannot act as a policy, it can still be assessed by evaluating the true reward performance of its direct outcome (i.e., the policy $\pi_\theta$). Thus, the optimization of the policy $\pi_\theta$ and the shaping weight function $z_\phi$ forms a bi-level optimization problem, which can be formally defined as

$$\max_\phi \ \mathbb{E}_{s\sim\rho^\pi, a\sim\pi_\theta}\left[r(s,a)\right]$$
$$\text{s.t. } \phi \in \Phi$$
$$\theta = \arg\max_{\theta'} \ \mathbb{E}_{s\sim\rho^\pi, a\sim\pi_{\theta'}}\left[r(s,a) + z_\phi(s,a)f(s,a)\right] \tag{2}$$
$$\text{s.t. } \theta' \in \Theta,$$

where $\Phi$ and $\Theta$ denote the parameter spaces of the shaping weight function and the policy, respectively. We call this problem *bi-level optimization of parameterized reward shaping* (BiPaRS, pronounced "bypass"). In this paper, we use notations like $x$ and $\tilde{x}$ to denote the variables with respect

to the original and modified MDPs, respectively. For example, we use $r$ to denote the true reward function and $\tilde{r}$ to denote the modified reward function.

## 3.2 Gradient Computation

The solution of the BiPaRS problem can be computed using a simple alternating optimization method. That is, to fix $z_\phi$ or $\pi_\theta$ and optimize the other alternately. Given the shaping weight function $z_\phi$, the lower level of BiPaRS is a standard policy optimization problem. Thus, the gradient of the expected accumulative modified reward $\tilde{J}$ with respect to the policy parameter $\theta$ is

$$\nabla_\theta \tilde{J}(\pi_\theta) = \mathbb{E}_{s \sim \rho^\pi, a \sim \pi_\theta} \left[ \nabla_\theta \log \pi_\theta(s, a) \tilde{Q}(s, a) \right]. \tag{3}$$

Here, $\tilde{Q}$ is the state-action value function of the current policy in the modified MDP $\tilde{\mathcal{M}} = \langle \mathcal{S}, \mathcal{A}, P, \tilde{r}, p_0, \gamma \rangle$. For the upper-level optimization, the following theorem is the basis for computing the gradient of $J(z_\phi)$ with respect to the variable $\phi$.

**Theorem 1.** *Given the shaping weight function $z_\phi$ and the stochastic policy $\pi_\theta$ of the agent in the upper level of the BiPaRS problem (Equation (2)), the gradient of the objective function $J(z_\phi)$ with respect to the variable $\phi$ is*

$$\nabla_\phi J(z_\phi) = \mathbb{E}_{s \sim \rho^\pi, a \sim \pi_\theta} \left[ \nabla_\phi \log \pi_\theta(s, a) Q^\pi(s, a) \right], \tag{4}$$

*where $Q^\pi$ is the state-action value function of $\pi_\theta$ in the original MDP.*

The complete proof is given in Appendix. Note that the theorem is based on the assumption that the gradient of $\pi_\theta$ with $\phi$ exists, which is reasonable because in the upper-level optimization, the given policy $\pi_\theta$ is the direct result of applying $z_\phi$ to reward shaping and in turn $\pi_\theta$ can be treated as an implicit function of $z_\phi$. However, even with this theorem, we still cannot get the accurate value of $\nabla_\phi J(z_\phi)$ because the gradient of the policy $\pi_\theta$ with respect to $\phi$ cannot be directly computed. In the next section, we will show how to approximate $\nabla_\phi \pi_\theta(s, a)$.

## 4 Gradient Approximation

### 4.1 Explicit Mapping

The idea of our first method for approximating $\nabla_\phi \pi_\theta(s, a)$ is to establish an explicit mapping from the shaping weight function $z_\phi$ to the policy $\pi_\theta$. Specifically, we redefine the agent's policy as a hyper policy $\pi_\theta : \mathcal{S}_z \times \mathcal{A} \to [0, 1]$ which can directly react to different shaping weights. Here, $\mathcal{S}_z = \{(s, z_\phi(s)) | \forall s \in \mathcal{S}\}$ is the extended state space and $z_\phi(s)$ is the shaping weight or shaping weight vector (e.g., $z_\phi(s) = (z_\phi(s, a_1), ..., z_\phi(s, a_{|\mathcal{A}|}))$) corresponding to state $s$. As $z_\phi$ is an input of the policy $\pi_\theta$, according to the chain rule we have $\nabla_\phi \pi_\theta(s, a, z_\phi(s)) = \nabla_z \pi_\theta(s, a, z_\phi(s)) \nabla_\phi z_\phi(s)$ and correspondingly the gradient of the upper-level objective function $J(z_\phi)$ with respect to $\phi$ is

$$\nabla_\phi J(z_\phi) = \mathbb{E}_{s \sim \rho^\pi, a \sim \pi_\theta} \left[ \nabla_z \log \pi_\theta(s, a, z)|_{z=z_\phi(s)} \nabla_\phi z_\phi(s) Q^\pi(s, a) \right]. \tag{5}$$

We call the first gradient approximating method *explicit mapping* (EM). Now we explain why the extension of the input space of $\pi_\theta$ is reasonable. For the lower-level optimization, having $z_\phi$ as the input of $\pi_\theta$ redefines the modified MDP $\tilde{\mathcal{M}} = \langle \mathcal{S}, \mathcal{A}, P, \tilde{r}, p_0, \gamma \rangle$ as a new MDP $\tilde{\mathcal{M}}_z = \langle \mathcal{S}_z, \mathcal{A}, P_z, \tilde{r}_z, p_z, \gamma \rangle$, where $\tilde{r}_z$, $P_z$ and $p_z$ also include $z_\phi$ in the input space and provide the same rewards, state transitions, and initial state probabilities as $\tilde{r}$, $P$, and $p_0$, respectively. Given the shaping weight function $z_\phi$, the one-to-one correspondence between the states in $\mathcal{S}$ and $\mathcal{S}_z$ indicates that the two MDPs $\tilde{\mathcal{M}}$ and $\tilde{\mathcal{M}}_z$ are equivalent.

### 4.2 Meta-Gradient Learning

Essentially, the policy $\pi_\theta$ and the shaping weight function $z_\phi$ are related because that their parameters $\theta$ and $\phi$ are related. The idea of our second method is to approximate the meta-gradient $\nabla_\phi \theta$ so that $\nabla_\phi \pi_\theta(s, a)$ can be computed as $\nabla_\theta \pi_\theta(s, a) \nabla_\phi \theta$. Given $\phi$, let $\theta$ and $\theta'$ denote the policy parameters before and after one round of low-level optimization, respectively. Without loss of generality,

we assume that a batch of $N$ $(N > 0)$ samples $\mathcal{B} = \{(s_i, a_i)|i = 1, ..., N\}$ is used for update $\theta$. According to Equation (3), we have

$$\theta' = \theta + \alpha \sum_{i=1}^{N} \nabla_\theta \log \pi_\theta(s_i, a_i) \tilde{Q}(s_i, a_i), \tag{6}$$

where $\alpha$ is the learning rate. The updated policy $\pi_{\theta'}$ then will be used for updating $\phi$ in the subsequent round of upper-level optimization, which involves the computation of $\nabla_\phi \theta'$. For any state-action pair $(s, a)$, we simplify the notation $\nabla_\theta \log \pi_\theta(s, a)$ as $g_\theta(s, a)$. Thus, the gradient of $\theta'$ with respect to $\phi$ can be computed as

$$\nabla_\phi \theta' = \nabla_\phi \big(\theta + \alpha \sum_{i=1}^{N} g_\theta(s_i, a_i) \tilde{Q}(s_i, a_i)\big) = \alpha \sum_{i=1}^{N} g_\theta(s_i, a_i)^\top \nabla_\phi \tilde{Q}(s_i, a_i). \tag{7}$$

We treat $\theta$ as a constant with respect to $\phi$ here because $\theta$ is the result of the last round of low-level optimization rather than the result of applying $z_\phi$ in this round. Now the problem is to compute $\nabla_\phi \tilde{Q}(s_i, a_i)$. Note that the state-action value function $\tilde{Q}$ can be replaced by any of its unbiased estimations such as the Monte Carlo return. For any sample $i$ in the batch $\mathcal{B}$, let $\tau_i = (s_i^0, a_i^0, \tilde{r}_i^0, s_i^1, a_i^1, \tilde{r}_i^1, ...)$ denote the sampled trajectory starting from $(s_i, a_i)$, where $(s_i^0, a_i^0)$ is $(s_i, a_i)$. The modified reward $\tilde{r}_i^t$ at each step $t$ of $\tau_i$ can be further denoted as $\tilde{r}_i^t = r_i^t + z_\phi(s_i^t, a_i^t) f(s_i^t, a_i^t)$, where $r_i^t$ is the sampled true reward. Therefore, $\nabla_\phi \theta'$ can be approximated as

$$\nabla_\phi \theta' \approx \alpha \sum_{i=1}^{N} g_\theta(s_i, a_i)^\top \nabla_\phi \sum_{t=0}^{|\tau_i|-1} \gamma^t \big(r_i^t + z_\phi(s_i^t, a_i^t) f(s_i^t, a_i^t)\big)$$

$$= \alpha \sum_{i=1}^{N} g_\theta(s_i, a_i)^\top \sum_{t=0}^{|\tau_i|-1} \gamma^t f(s_i^t, a_i^t) \nabla_\phi z_\phi(s_i^t, a_i^t). \tag{8}$$

### 4.3 Incremental Meta-Gradient Learning

The third method for gradient approximation is a generalized version of the meta-gradient learning (MGL) method. Recall that in Equation (7), the old policy parameter $\theta$ is treated as a constant with respect to $\phi$. However, $\theta$ can be also considered as a non-constant with respect to $\phi$ because in the last round of upper-level optimization, $\phi$ is updated according to the true rewards sampled by the old policy $\pi_\theta$. Furthermore, as the parameters of the policy and the shaping weight function are updated incrementally round by round, both of them actually are related to all previous versions of the other. Therefore, we can rewrite Equation (7) as

$$\nabla_\phi \theta' = \nabla_\phi \theta + \alpha \sum_{i=1}^{N} \nabla_\phi g_\theta(s_i, a_i)^\top \tilde{Q}(s_i, a_i) + \alpha \sum_{i=1}^{N} g_\theta(s_i, a_i)^\top \nabla_\phi \tilde{Q}(s_i, a_i)$$

$$= \nabla_\phi \theta + \alpha \sum_{i=1}^{N} \big(\nabla_\theta g_\theta(s_i, a_i)^\top \nabla_\phi \theta\big) \tilde{Q}(s_i, a_i) + \alpha \sum_{i=1}^{N} g_\theta(s_i, a_i)^\top \nabla_\phi \tilde{Q}(s_i, a_i) \tag{9}$$

$$= \big(I_n + \alpha \sum_{i=1}^{N} \tilde{Q}(s_i, a_i) \nabla_\theta g_\theta(s_i, a_i)^\top\big) \nabla_\phi \theta + \alpha \sum_{i=1}^{N} g_\theta(s_i, a_i)^\top \nabla_\phi \tilde{Q}(s_i, a_i),$$

where $I_n$ is a $n$-order identity matrix, and $n$ is the number of parameters in $\theta$. In practice, we can initialize the gradient of the policy parameter with respect to $\phi$ to a zero matrix and update it according to Equation (9) iteratively. Due to this incremental computing manner, we call the new method *incremental meta-gradient learning* (IMGL). Actually, the above three methods make different tradeoffs between the accuracy of gradient computation and computational complexity. In Appendix, we will provide complexity analysis of these methods and the details of the full algorithm.

## 5 Experiments

We conduct three groups of experiments. The first one is conducted in *cartpole* to verify that our methods can improve the performance of a learning algorithm with beneficial shaping rewards. The

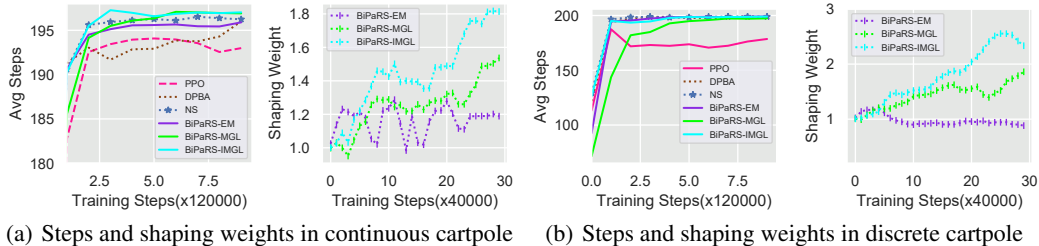

(a) Steps and shaping weights in continuous cartpole    (b) Steps and shaping weights in discrete cartpole

Figure 1: The average step results and the shaping weights of our algorithms in the cartpole tasks

second one is conducted in *MuJoCo* to further test our algorithms in more complex problems. The last one is an adaptability test which shows that our methods can learn good policies and correct shaping weights even when the provided shaping reward function is harmful.

## 5.1 Sparse-Reward Cartpole

In *cartpole*, the agent should apply a force to the pole to keep it from falling. The agent will receive a reward $-1$ from the environment if the episode ends with the falling of the pole. In other cases, the true reward is zero. The shaping reward for the agent is $0.1$ if the force applied to the cart and the deviation angle of the pole have the same sign. Otherwise, the shaping reward is zero. In short, such a shaping reward function encourages the actions which make the deviation angle smaller and definitely is beneficial. We adopt the PPO algorithm [16] as the base learner and test our algorithms in both continuous and discrete action settings. We evaluate the three versions of our algorithm, namely BiPaRS-EM, BiPaRS-MGL, and BiPaRS-IMGL and compare them with the naive shaping (NS) method which directly adds the shaping rewards to the original ones, and the dynamic potential-based advice (DPBA) method [6] which transforms an arbitrary reward into a potential value. The details of the *cartpole* problem and the algorithm settings are given in Appendix.

**Test Setting:** The test of each method contains $1,200,000$ training steps. During the training process, a 20-episode evaluation is conducted every $4,000$ steps and we record the average steps per episode (ASPE) performance of the tested method at each evaluation point. The ASPE performance stands for how long the agent can keep the pole from falling. The maximal length of an episode is 200, which means that the highest ASPE value that can be achieved is 200. To investigate how our algorithms adjust the shaping weights, we also record the average shaping weights of the experienced states or state-action pairs every $4,000$ steps. The shaping weights of our methods are initialized to 1.

**Results:** The results of all tests are averaged over 20 independent runs using different random seeds and are shown in Figure 1. With the provided shaping reward function, all these methods can improve the learning performance of the PPO algorithm (the left columns in Figs 1(a) and 1(b)). In the continuous-action cartpole task, the performance gap between PPO and the shaping methods is small. In the discrete-action cartpole task, PPO only converges to 170, but with the shaping methods it almost achieves the highest ASPE value 200. By investigating the shaping weight curves in Figure 1, it can be found that our algorithms successfully recognize the positive utility of the given shaping reward function, since all of them keep outputting positive shaping weights during the learning process. For example, in Figure 1(a), the average shaping weights of the BiPaRS-MGL and BiPaRS-IMGL algorithms start from $1.0$ and finally reach $1.5$ and $1.8$, respectively. The BiPaRS-EM algorithm also learns higher shaping weights than the initial values.

## 5.2 MuJoCo

We choose five MuJoCo tasks *Swimmer-v2*, *Hopper-v2*, *Humanoid-v2*, *Walker2d-v2*, and *HalfCheetah-v2* from OpenAI Gym-v1 to test our algorithms. In each task, the agent is a robot composed of several joints and should decide the amount of torque to apply to each joint in each step. The true reward function is the one predefined in Gym. The shaping reward function is adopted from a recent paper [22] which studies policy optimization with reward constraints and proposes the reward constrained policy optimization (RCPO) algorithm. Specifically, for each state-action pair

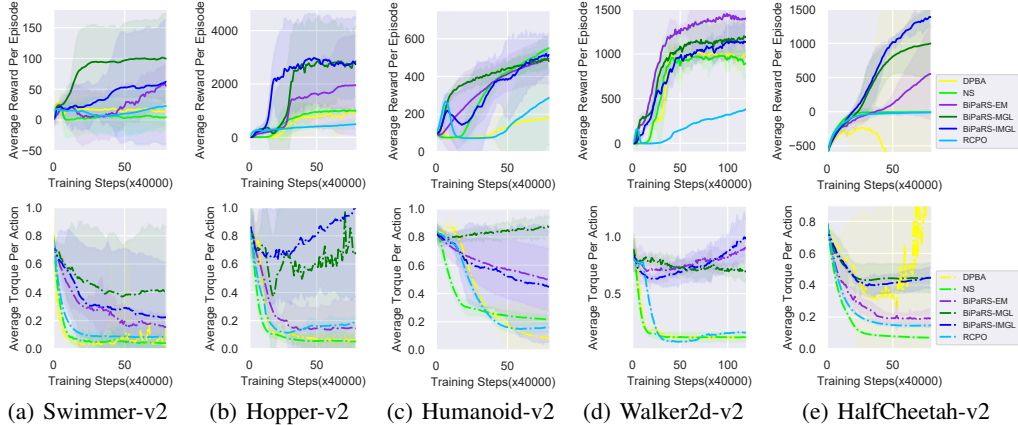

Figure 2: Results of the MuJoCo experiment. The shaded areas are $95\%$ confidence intervals

$(s, a)$, we define its shaping reward $f(s, a) = w(0.25 - \frac{1}{L}\sum_{i=1}^{L}|a_i|)$, where $L$ is the number of joints, $a_i$ is the torque applied to joint $i$, and $w$ is a task-specific weight (given in Appendix) which makes $f(s, a)$ have the same scale as the true reward. Concisely, $f$ requires the average torque amount of each action to be smaller than $0.25$, which most probably can result in a sub-optimal policy [22]. Although such shaping reward function is originally used as constraints for achieving safety, we only care about whether our methods can benefit the maximization of the true rewards and how they will do if the shaping rewards are in conflict with the true rewards.

**Test Setting:** We adopt PPO as the base learner and still compare our shaping methods with the NS and DPBA methods. The RCPO algorithm [22] is also adopted as a baseline. The tests in *Walker2d-v2* contain $4,800,000$ training steps and the other tests contain $3,200,000$. A 20-episode evaluation is conducted every 4,000 training steps. We record the average reward per episode (ARPE) performance and the average torque amount of the agent's actions at each evaluation point. The maximal length of an episode is 1000.

**Results:** The results of the MuJoCo tests are averaged over 10 runs with different random seeds and are shown in Figure 2. Our algorithms adapt well to the given shaping reward function in each test. For example, in the *Hopper-v2* task (Figure 2(b)), BiPaRS-MGL and BiPaRS-IMGL are far better than the other methods. The BiPaRS-EM method performs slightly worse, but is still better than the baseline methods. The torque amount curves of the tested algorithms are consistent with their reward curves. In some tasks such as *Swimmer-v2* and *HalfCheetah-v2*, the BiPaRS methods slow down the decaying speed of the agent's torque amount. In the other tasks such as *Hopper-v2* and *Walker2d-v2*, our methods even raise the torque amount values. Interestingly, in Figure 2(b) we can find a v-shaped segment on each of the torque curves of BiPaRS-MGL and BiPaRS-IMGL. This indicates that the two algorithms try to reduce the torque amount of actions at first, but do the opposite after realizing that a small torque amount is unbeneficial. The algorithm performance in our *MuJoCo* test may not match the state-of-the-art in the DRL domain since $f$ brings additional difficulty for policy optimization. It should also be noted that the goal of this test is to show the disadvantage of fully exploiting the given shaping rewards and our algorithms have the ability to avoid this.

## 5.3 Adaptability Test

Three additional tests are conducted in *cartpole* to further investigate the adaptability of our methods. In the first test, we adopt a harmful shaping reward function which gives the agent a reward $-0.1$ when the deviation angle of the pole becomes smaller. In the second test, the same shaping reward function is adopted, but our BiPaRS algorithms reuse the shaping weight functions learnt in the first test and only optimize the policy. The goal of the two tests is to show that our methods can identify the harmful shaping rewards and the learnt shaping weights are helpful. In the third test, we use shaping rewards randomly distributed in $[-1, 1]$ to investigate how our methods adapt to an arbitrary shaping reward function. The other settings are the same as the original *cartpole* experiment.

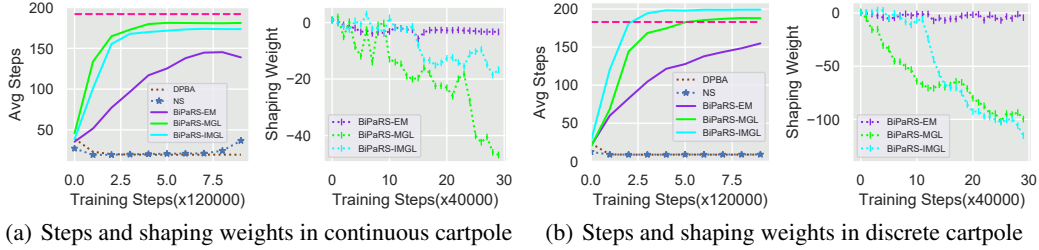

(a) Steps and shaping weights in continuous cartpole   (b) Steps and shaping weights in discrete cartpole

Figure 3: The average step and shaping weight curves of our algorithms in cartpole with a harmful shaping reward function. The magenta lines represent the values that PPO reaches in Figure 1

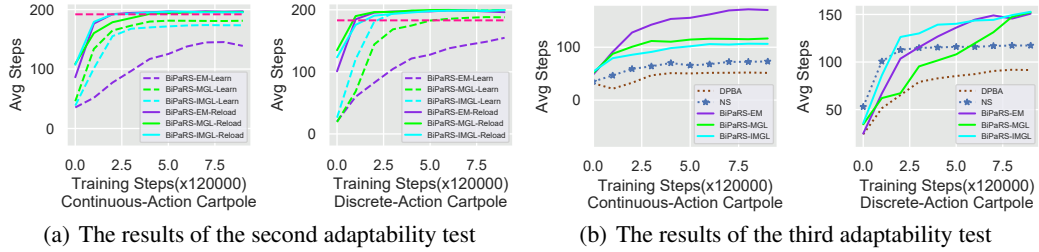

(a) The results of the second adaptability test   (b) The results of the third adaptability test

Figure 4: The results of the second adaptability test where the BiPaRS algorithms reload the learnt shaping weight functions and the third adaptability test where the shaping rewards are randomly distributed in $[-1, 1]$

The results of the first test are shown in Figure 3. The unbeneficial shaping rewards inevitably hurt the policy optimization process, so directly comparing the learning performance of PPO with the tested shaping methods is meaningless. Instead, in Figure 3 we plot the step values achieved by PPO in experiment 5.1 as reference (the magenta lines). We can find that the NS and DPBA methods perform poorly since their step curves stay at a low level in the whole training process. In contrast, our methods recognize that the shaping reward function is harmful and have made efforts to reduce its bad influence. For example, in Figure 3(a) BiPaRS-EM finally achieves a decent ASPE value 130, and the step curves of BiPaRS-MGL and BiPaRS-IMGL are very close to the reference line (around 180). In discrete-action *cartpole*, the two methods even achieve higher values than the reference line (Figure 3(b)). The negative shaping weights shown in right columns of Figures 3(a) and 3(b) indicate that our methods are trying to transform the harmful shaping rewards into beneficial ones.

We put the results of the second and third tests in Figures 4(a) and 4(b), respectively. Now examine Figure 4(a), where the dashed lines are the step curves of our methods in the first test. Obviously, the reloaded shaping weight functions enables the BiPaRS methods to learn much better and faster. Furthermore, the learnt policies are better than the policy learnt by PPO, which proves that the learnt shaping weights are correct and helpful. In the third adaptability test, our methods also perform better than the baseline methods. The BiPaRS-EM method achieves the highest step value (about 170) in the continuous-action cartpole, and all our three methods reach the highest step value 150 in the discrete-action cartpole. Note that the third test is no easier than the first one because an algorithm may get stuck with opposite shaping reward signals of similar state-action pairs.

## 5.4 Learning State-Dependent Shaping Weights

One may have noticed that the shaping reward functions adopted in the above experiments (except the random one in the third adaptability test) are either fully helpful or fully harmful, which means that a uniform shaping weight may also work for adaptive utilization of the shaping rewards. In fact, the shaping weights learnt by our methods differ slightly across the state-action space in the cartpole experiment in Section 5.1 and the first adaptability test in Section 5.3. Learning a state-action-independent shaping weight in the two tests seems sufficient. We implement the baseline method which replaces the shaping weight function $z_\phi$ with a single shaping weight and test it in the two experiments. It can be found from the corresponding results (Figures 5(a) and 5(b)) that the single-

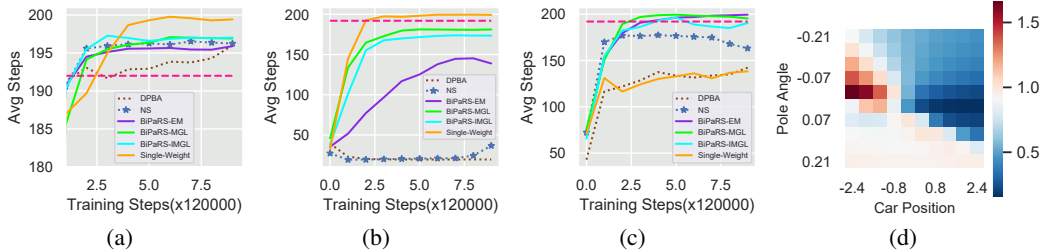

Figure 5: Results of the additional experiments. Figures 5(a) and 5(b) are the results of the single-weight baseline method in continuous-action carpole with beneficial and harmful shaping rewards, respectively. Figure 5(c) is the result of the experiment with beneficial and harmful shaping rewards in different states, and Figure 5(d) is the heat map of BiPaRS-EM's shaping weights of $100$ states

weight method learns better and faster than all the other methods. Although our methods can identify the benefits or harmfulness of a given shaping reward function, it is still unclear whether they have the ability to learn a state-dependent or state-action-depenent shaping weight function. Therefore, we conduct an additional experiment in the continuous-action cartpole environment with half of the shaping rewards are helpful and the other half of the shaping rewards are harmful. Specifically, the shaping reward is $0.1$ if the action taken by the agent reduces the deviation angle when the pole inclines to the right. However, if the action of the agent reduces the deviation angle when the pole inclines to the left, the corresponding shaping reward is $-0.1$. The result is shown in Figure 5(c). It can be found that our methods perform the best and the single-weight baseline method cannot perform as well as in Figures 5(a) and 5(b). For better illustration, we also plot the shaping weights learnt by the BiPaRS-EM method across a subset of the state space (containing 100 states) as a heat map in Figure 5(d). Note that a state in the cartpole task is represented by a $4$-dimensional vector. For the $100$ states used for drawing the heat map, we fix the car velocity $(1.0)$ and pole velocity $(0.01)$ features, and only change the car position and pole angle features. From the heat map figure, we can see that as either of the two state feature changes, the corresponding shaping weight also changes considerably, which indicates that state-dedependent shaping weights are learnt.

## 6 Conclusions

In this work, we propose the *bi-level optimization of parameterized reward shaping* (BiPaRS) approach for adaptively utilizing a given shaping reward function. We formulate the utilization problem of shaping rewards, provide formal results for gradient computation, and propose three learning algorithms for solving this problem. The results in cartpole and MuJoCo tasks show that our algorithms can fully exploit beneficial shaping rewards, and meanwhile ignore unbeneficial shaping rewards or even transform them into beneficial ones.

## Broader Impact

Reward design is an important and difficult problem in real-world applications of reinforcement learning (RL). In many cases, researchers or algorithm engineers have some prior knowledge (such as rules and constraints) about the problem to be solved, but cannot represent the knowledge as numeric values very exactly. Improper reward settings may be exploited by an RL algorithm to obtain higher rewards, but with unexpected behaviors learnt. This paper provides an adaptive approach of reward shaping to avoid the repeated and tedious tuning of rewards in RL applications (e.g., video games). A direct impact of our paper is that researchers or algorithm engineers can be liberated from hard work of reward tuning. The shaping weights learnt by our methods indicate the quality of the designed rewards, and thus can help the designers to better understand the problems. Our paper also proposes a general principle for utilizing prior knowledge in the machine learning domain, namely trying to get rid of human cognitive error when the qualitative or rule-based prior knowledge is transformed into numeric values to help with learning.

## Acknowledgments and Disclosure of Funding

We thank the reviewers for the valuable comments and suggestions. Weixun Wang and Jianye Hao are supported by the National Natural Science Foundation of China (Grant Nos.: 61702362, U1836214), Special Program of Artificial Intelligence and Special Program of Artificial Intelligence of Tianjin Municipal Science and Technology Commission (No.: 569 17ZXRGGX00150). Yixiang Wang and Feng Wu are supported in part by the National Natural Science Foundation of China (Grant No. U1613216, Grant No. 61603368).

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
