[Supplementary Material · appendix.pdf]

# A  Complexity and Algorithm

## A.1  Complexity Analysis

In this section, we provide the complexity analysis of the gradient approximation methods proposed in Section 4 and show the full algorithm for solving the bi-level optimization of parameterized reward shaping (BiPaRS) problem. We denote the number of parameters of the policy function $\pi_\theta$ by $n$ and the number of parameters of the shaping weight function $z_\phi$ by $m$, respectively.

**Explicit Mapping**: The explicit mapping (EM) method includes the shaping weight function $z_\phi$ as an input of $\pi_\theta$ and approximately computes $\nabla_\phi \log \pi_\theta(s, a)$ as $\nabla_z \log \pi_\theta(s, a, z)|_{z=z_\phi(s)} \nabla_\phi z_\phi(s)$. Therefore, the computational complexity of the EM method is $\mathcal{O}(m)$.

**Meta-Gradient Learning**: Given the shaping weight function parameter $\phi$, let $\theta$ and $\theta'$ denote the policy parameters before and after one round of low-level optimization, and assume that a batch of $N$ ($N > 0$) samples $\mathcal{B} = \{(s_i, a_i)|i = 1, ..., N\}$ is used for updating $\theta$. The meta-gradient learning (MGL) method computes $\nabla_\phi \theta'$ as

$$
\begin{aligned}
\nabla_\phi \theta' &\approx \alpha \sum_{i=1}^{N} g_\theta(s_i, a_i)^\top \nabla_\phi \sum_{t=0}^{|\tau_i|-1} \gamma^t \big( r_i^t + z_\phi(s_i^t, a_i^t) f(s_i^t, a_i^t) \big) \\
&= \alpha \sum_{i=1}^{N} g_\theta(s_i, a_i)^\top \sum_{t=0}^{|\tau_i|-1} \gamma^t f(s_i^t, a_i^t) \nabla_\phi z_\phi(s_i^t, a_i^t).
\end{aligned}
\tag{A.1}
$$

It seems that the computational complexity and space complexity of the (MGL) method are $\mathcal{O}(Nnm)$. However, we can reduce both of them to $\mathcal{O}(N(n+m))$. Note that Equation (A.1) should be integrated with Theorem 1 for computing the gradient of the expected true reward $J(z_\phi)$ with respect to $\phi$ in the upper-level optimization. Without loss of generality, we assume only one state-action pair $(s, a)$ is used to update $\phi$ under the new policy $\theta'$. By substituting Equation (A.1) into Theorem 1, we have

$$
\begin{aligned}
\nabla_\phi J(z_\phi) &\approx \nabla_\phi \log \pi_{\theta'}(s, a) Q(s, a) \\
&= \nabla_{\theta'} \log \pi_{\theta'}(s, a) \nabla_\phi \theta' Q(s, a) \\
&\approx \alpha \nabla_{\theta'} \log \pi_{\theta'}(s, a) \Big( \sum_{i=1}^{N} g_\theta(s_i, a_i)^\top \sum_{t=0}^{|\tau_i|-1} \gamma^t f(s_i^t, a_i^t) \nabla_\phi z_\phi(s_i^t, a_i^t) \Big) Q(s, a),
\end{aligned}
\tag{A.2}
$$

where $Q(s, a)$ is the state-action value function under the policy $\pi_{\theta'}$. Actually, we can change the computing order of Equation (A.2) and compute it as

$$
\nabla_\phi J(z_\phi) \approx \alpha \sum_{i=1}^{N} \big( \nabla_{\theta'} \log \pi_{\theta'}(s, a) Q(s, a) g_\theta(s_i, a_i)^\top \big) \sum_{t=0}^{|\tau_i|-1} \gamma^t f(s_i^t, a_i^t) \nabla_\phi z_\phi(s_i^t, a_i^t).
\tag{A.3}
$$

For each $i$ in the sum loop of this equation, computing the term $\nabla_{\theta'} \log \pi_{\theta'}(s, a) Q(s, a) g_\theta(s_i, a_i)^\top$ costs $\mathcal{O}(n)$ time and space, computing the term $\sum_{t=0}^{|\tau_i|-1} \gamma^t f(s_i^t, a_i^t) \nabla_\phi z_\phi(s_i^t, a_i^t)$ costs $\mathcal{O}(m)$ time and space (we treat $|\tau_i|$ as a constant), and computing the product of these two terms also costs $\mathcal{O}(m)$ time and space. Therefore, the space complexity and computational complexity for computing Equation (A.3) are $\mathcal{O}(N(n+m))$.

**Incremental Meta-Gradient Learning:** The incremental meta-gradient learning (IMGL) is a generalized version of the MGL method. We still let $\theta$ and $\theta'$ denote the policy parameters before and after one round of low-level optimization, and assume that a batch of $N$ ($N > 0$) samples $\mathcal{B} = \{(s_i, a_i)|i = 1, ..., N\}$ is used to update $\theta$. The IMGL method computes $\nabla_\phi \theta'$ as

$$
\begin{aligned}
\nabla_\phi \theta' &= \nabla_\phi \theta + \alpha \sum_{i=1}^{N} \nabla_\phi g_\theta(s_i, a_i)^\top \tilde{Q}(s_i, a_i) + \alpha \sum_{i=1}^{N} g_\theta(s_i, a_i)^\top \nabla_\phi \tilde{Q}(s_i, a_i) \\
&= \big( I_n + \alpha \sum_{i=1}^{N} \tilde{Q}(s_i, a_i) \nabla_\theta g_\theta(s_i, a_i)^\top \big) \nabla_\phi \theta + \alpha \sum_{i=1}^{N} g_\theta(s_i, a_i)^\top \nabla_\phi \tilde{Q}(s_i, a_i),
\end{aligned}
\tag{A.4}
$$

Figure 3: An overview of the BiPaRS framework

where $I_n$ is a $n$-order identity matrix and $\tilde{Q}$ denotes the state-action value function in the modified MDP. Compared with the EM and MGL methods, IMGL is much more computationally expensive since the term $\alpha \sum_{i=1}^{N} \tilde{Q}(s_i, a_i) \nabla_\theta g_\theta(s_i, a_i)^\top$ involves the computation of the Hessian matrix $\nabla_\theta g_\theta(s_i, a_i)^\top$ for each $i$, which has $\mathcal{O}(Nn^3)$ computational complexity. However, there are several ways for reducing the high computational cost of IMGL. Firstly, the Hessian matrix $\nabla_\theta g_\theta(s_i, a_i)^\top$ can be approximated using outer product of gradients (OPG) estimate. Secondly, for simple problems, we can use a small policy model with a few parameters. Lastly, we can even omit the second-order term $\alpha \sum_{i=1}^{N} \tilde{Q}(s_i, a_i) \nabla_\theta g_\theta(s_i, a_i)^\top$ to get another approximation of $\nabla_\phi \theta'$.

## A.2 The Full Algorithm

The main workflow of the BiPaRS framework can be illustrated by Figure 3. In this figure, the left column shows the training loop of the agent's policy, where the expert knowledge (i.e., shaping reward) is integrated with the samples generated from the agent-environment interaction. The central column shows the difference between the parameterized reward shaping and the traditional reward shaping methods, namely the shaping weight function $z_\phi$ for adaptive utilization of the shaping reward function $f$. The right column of the figure intuitively shows the alternating optimization process of the policy and shaping weight function.

Now we summarize all the methods proposed in our paper into the general learning algorithm in Algorithm 1. This algorithm actually corresponds to three specifi learning algorithms which adopt the explicit mapping (EM), meta-gradient learning (MGL), and incremental meta-gradient learning (IMGL) methods for gradient approximation, respectively. As shown in the algorithm table, the policy parameter $\theta$ and the parameter of the shaping weight function $\phi$ are optimized iteratively. At each iteration $t$, $\theta$ is firstly updated according to the shaping rewards (lines 5 to 14), which are weighted by the shaping weight function $z_\phi$. If MGL or IMGL is chosen as the method for gradient approximation, then the meta-gradient $\nabla_\phi \theta$, which is denoted by the variable $h$, will be computed at the same time (lines 9 to 14), where $\nabla_\theta \log \pi_\theta(s, a)$ is simplified as $g_\theta(s, a)$. After updating $\theta$, $\phi$ is updated based on the true rewards sampled by the new policy $\pi_\theta$ and the approximated gradient of $\pi_\theta$ with respect to $\phi$ (lines 15 to 22). For the EM method (line 21), we do not explicitly represent $\pi_\theta$ as a function of $z_\phi$ in order to keep the consistency of notations. We also omit some details in Algorithm 1, such as the learning of the two value functions $Q$ and $\tilde{Q}$, and the computation of the gradient $\nabla_\phi \tilde{Q}(s, a)$.

# B Theorem Proof

This section gives the proof of Theorem 1. We make the following assumptions.

**Assumption 1.** *Let $\mathcal{M} = \langle \mathcal{S}, \mathcal{A}, P, r, p_0, \gamma \rangle$ denote the original MDP, $\pi_\theta$ be the policy, and $z_\phi$ be the shaping weight function in the BiPaRS problem, repsectively. We assume that $P(s, a, s')$, $r(s, a)$, $p_0(s)$ are continuous w.r.t. the variables $s$, $a$, and $s'$, and $\pi_\theta(s, a)$ are continuous w.r.t. $s$, $a$, $\theta$, and $\phi$.*

**Assumption 2.** *For the MDP $\mathcal{M} = \langle \mathcal{S}, \mathcal{A}, P, r, p_0, \gamma \rangle$, there exists a real number $b$ such that $r(s, a) < b$, $\forall (s, a)$.*

**Theorem 1.** *Given the shaping weight function $z_\phi$ and the stochastic policy $\pi_\theta$ of the agent in the upper level of the BiPaRS problem (Equation (2) in the paper), the gradient of the objective function*

---

**Algorithm 1:** Bilevel Optimization of Parameterized Reward Shaping (BiPaRS)

---
**Input:** Learning rates $\alpha_\theta$ and $\alpha_\phi$
1 Initialize the policy parameter $\theta$ and the shaping weight function parameter $\phi$;
2 Initialize the true and shaping value functions $Q$ and $\tilde{Q}$;
3 Initialize the meta-gradient $h$ to a zero matrix;
4 **for** $t = 1, 2, ..., $ **do**
5      Run policy $\pi_\theta$ in the modified MDP with $z_\phi$;
6      $\mathcal{T}' \leftarrow$ the set of sampled experiences;
7      Update $\tilde{Q}$ using samples from $\mathcal{T}'$;
8      $\theta' \leftarrow \theta + \alpha_\theta \sum_{(s,a)\sim\mathcal{T}'} \nabla_\theta \log \pi_\theta(s,a)\tilde{Q}(s,a)$;
9      **if** *MGL is used* **then**
10          $h \leftarrow \alpha_\theta \sum_{(s,a)\sim\mathcal{T}'} g_\theta(s,a)^\top \nabla_\phi \tilde{Q}(s,a)$;
11      **else if** *IMGL is used* **then**
12          $h_1 \leftarrow \big(\sum_{(s,a)\sim\mathcal{T}'} \nabla_\theta g_\theta(s,a)^\top \tilde{Q}(s,a)\big)h$;
13          $h_2 \leftarrow \sum_{(s,a)\sim\mathcal{T}'} g_\theta(s,a)\nabla_\phi \tilde{Q}(s,a)$;
14          $h \leftarrow h + \alpha_\theta(h_1 + h_2)$;
15      Run policy $\pi_{\theta'}$ in the original MDP;
16      $\mathcal{T} \leftarrow$ the set of sampled experiences;
17      Update $Q$ using samples from $\mathcal{T}$;
18      **if** *MGL or IMGL is used* **then**
19          $\Delta\phi \leftarrow \sum_{(s,a)\sim\mathcal{T}} \nabla_{\theta'} \log \pi_{\theta'}(s,a)Q(s,a)h$;
20      **else**
21          $\Delta\phi \leftarrow \sum_{(s,a)\sim\mathcal{T}} \nabla_z \log \pi_{\theta'}(s,a)\nabla_\phi z_\phi(s)Q(s,a)$;
22      $\phi \leftarrow \phi + \alpha_\phi \Delta\phi$;
23      $\theta \leftarrow \theta'$;

---

69    $J(z_\phi)$ *with respect to the variable $\phi$ is*

$$\nabla_\phi J(z_\phi) = \mathbb{E}_{s\sim\rho^\pi, a\sim\pi_\theta}\big[\nabla_\phi \log \pi_\theta(s,a)Q^\pi(s,a)\big], \tag{B.5}$$

70    *where $Q^\pi$ is the state-action value function of $\pi_\theta$ in the original MDP.*

71    *Proof.* Our proof is similar to that of the stochastic policy gradient theorem (Sutton et al. (1999))
72    and the deterministic policy gradient theorem (Silver et al. (2014)). Given a stochastic policy $\pi_\theta$,
73    let $V^\pi$ and $Q^\pi$ denote the state value function and state-action value function of $\pi$ in the original
74    MDP $\mathcal{M} = \langle \mathcal{S}, \mathcal{A}, P, r, p_0, \gamma \rangle$, respectively. Note that Assumption 1 implies that $V^\pi$ and $Q^\pi$
75    are continuous functions of $\phi$ and Assumption 2 guarantees that $V^\pi(s)$, $Q^\pi(s,a)$, $\nabla_\phi V^\pi(s)$ and
76    $\nabla_\phi Q^\pi(s,a)$ are bounded for any $s \in \mathcal{S}$ and any $a \in \mathcal{A}$. In our proof, the two assumptions are
77    necessary to exchange integrals and derivatives, and the integration orders.

78    Obviously, for any state $s$ we have

$$V^\pi(s) = \int_{\mathcal{A}} \pi_\theta(s,a)Q^\pi(s,a)\mathrm{d}a.$$

79    Therefore, the gradient of $V^\pi(s)$ with respect to the shaping weight function parameter $\phi$ is

$$\begin{aligned}
\nabla_\phi V^\pi(s) &= \nabla_\phi\Big(\int_{\mathcal{A}} \pi_\theta(s,a)Q^\pi(s,a)\mathrm{d}a\Big) \\
&= \int_{\mathcal{A}}\Big(\nabla_\phi \pi_\theta(s,a)Q^\pi(s,a) + \pi_\theta(s,a)\nabla_\phi Q^\pi(s,a)\Big)\mathrm{d}a.
\end{aligned}$$

80    The above equation is an application of the Leibniz integral rule to exchange the orders of derivative
81    and integral, which requires Assumption 1.

82  By further expanding the term $\nabla_\phi Q^\pi(s, a)$, we can obtain

$$
\begin{aligned}
\nabla_\phi V^\pi(s) &= \int_\mathcal{A} \Big( \nabla_\phi \pi_\theta(s, a) Q^\pi(s, a) + \pi_\theta(s, a) \nabla_\phi Q^\pi(s, a) \Big) \mathrm{d}a \\
&= \int_\mathcal{A} \Big( \nabla_\phi \pi_\theta(s, a) Q^\pi(s, a) + \pi_\theta(s, a) \nabla_\phi \big( r(s, a) + \gamma \int_\mathcal{S} P(s, a, s') V^\pi(s') \mathrm{d}s' \big) \Big) \mathrm{d}a \\
&= \int_\mathcal{A} \nabla_\phi \pi_\theta(s, a) Q^\pi(s, a) \mathrm{d}a + \int_\mathcal{A} \pi_\theta(s, a) \int_\mathcal{S} \gamma P(s, a, s') \nabla_\phi V^\pi(s') \mathrm{d}s' \mathrm{d}a \\
&= \int_\mathcal{A} \nabla_\phi \pi_\theta(s, a) Q^\pi(s, a) \mathrm{d}a + \int_\mathcal{S} \int_\mathcal{A} \gamma \pi_\theta(s, a) P(s, a, s') \mathrm{d}a \nabla_\phi V^\pi(s') \mathrm{d}s' \\
&= \int_\mathcal{A} \nabla_\phi \pi_\theta(s, a) Q^\pi(s, a) \mathrm{d}a + \int_\mathcal{S} \gamma p(s \to s', 1, \pi_\theta) \nabla_\phi V^\pi(s') \mathrm{d}s',
\end{aligned}
$$

83  where $p(s' \to s, t, \pi_\theta)$ is the probability that state $s$ is visited after $t$ steps from state $s'$ under the
84  policy $\pi_\theta$. In the above derivation, the first step is an expansion of the Bellman equation. The second
85  step is by exchanging the order of derivative and integral. The third step exchanges the order of
86  integration by using Fubini's Theorem, which requires our assumptions. The last step is according to
87  the definition of $p$. By expanding $\nabla_\phi V^\pi(s')$ in the same way, we can get

$$
\begin{aligned}
\nabla_\phi V^\pi(s) &= \int_\mathcal{A} \nabla_\phi \pi_\theta(s, a) Q^\pi(s, a) \mathrm{d}a + \int_\mathcal{S} \gamma p(s \to s', 1, \pi_\theta) \nabla_\phi \Big( \int_\mathcal{A} \pi_\theta(s', a') Q^\pi(s', a') \mathrm{d}a' \Big) \mathrm{d}s' \\
&= \int_\mathcal{A} \nabla_\phi \pi_\theta(s, a) Q^\pi(s, a) \mathrm{d}a + \int_\mathcal{S} \gamma p(s \to s', 1, \pi_\theta) \int_\mathcal{A} \nabla_\phi \pi_\theta(s', a') Q^\pi(s', a') \mathrm{d}a' \mathrm{d}s' \\
&\quad + \int_\mathcal{S} \gamma p(s \to s', 1, \pi_\theta) \int_\mathcal{A} \pi_\theta(s', a') \nabla_\phi Q^\pi(s', a') \mathrm{d}a' \mathrm{d}s' \\
&= \int_\mathcal{A} \nabla_\phi \pi_\theta(s, a) Q^\pi(s, a) \mathrm{d}a + \int_\mathcal{S} \gamma p(s \to s', 1, \pi_\theta) \int_\mathcal{A} \nabla_\phi \pi_\theta(s', a') Q^\pi(s', a') \mathrm{d}a' \mathrm{d}s' \\
&\quad + \int_\mathcal{S} \gamma p(s \to s', 1, \pi_\theta) \int_\mathcal{S} \gamma p(s' \to s'', 1, \pi_\theta) \nabla_\phi V^\pi(s'') \mathrm{d}s'' \mathrm{d}s' \\
&= \int_\mathcal{A} \nabla_\phi \pi_\theta(s, a) Q^\pi(s, a) \mathrm{d}a + \int_\mathcal{S} \gamma p(s \to s', 1, \pi_\theta) \int_\mathcal{A} \nabla_\phi \pi_\theta(s', a') Q^\pi(s', a') \mathrm{d}a' \mathrm{d}s' \\
&\quad + \int_\mathcal{S} \gamma p(s \to s', 2, \pi_\theta) \nabla_\phi V^\pi(s') \mathrm{d}s' \\
&= \cdots \\
&= \sum_{t=0}^\infty \int_\mathcal{S} \gamma^t p(s \to s', t, \pi_\theta) \int_\mathcal{A} \nabla_\phi \pi_\theta(s', a) Q^\pi(s', a) \mathrm{d}a \mathrm{d}s' \\
&= \int_\mathcal{S} \sum_{t=0}^\infty \gamma^t p(s \to s', t, \pi_\theta) \int_\mathcal{A} \nabla_\phi \pi_\theta(s', a) Q^\pi(s', a) \mathrm{d}a \mathrm{d}s'.
\end{aligned}
$$

88  Recall that the upper-level objective $J(z_\phi) = \int_\mathcal{S} \rho^\pi(s) \int_\mathcal{A} r(s, a) \mathrm{d}a \mathrm{d}s = \int_\mathcal{S} p_0(s) V^\pi(s) \mathrm{d}s$. There-
89  fore, we have

$$
\begin{aligned}
\nabla_\phi J(z_\phi) &= \int_\mathcal{S} p_0(s) \nabla_\phi V^\pi(s) \mathrm{d}s \\
&= \int_\mathcal{S} p_0(s) \int_\mathcal{S} \sum_{t=0}^\infty \gamma^t p(s \to s', t, \pi_\theta) \int_\mathcal{A} \nabla_\phi \pi_\theta(s', a) Q^\pi(s', a) \mathrm{d}a \mathrm{d}s' \mathrm{d}s \\
&= \int_\mathcal{S} p_0(s) \int_\mathcal{S} \sum_{t=0}^\infty \gamma^t p(s \to s', t, \pi_\theta) \mathrm{d}s \int_\mathcal{A} \nabla_\phi \pi_\theta(s', a) Q^\pi(s', a) \mathrm{d}a \mathrm{d}s' \\
&= \int_\mathcal{S} \rho^\pi(s') \int_\mathcal{A} \nabla_\phi \pi_\theta(s', a) Q^\pi(s', a) \mathrm{d}a \mathrm{d}s'.
\end{aligned}
$$

90  By using the log-derivative trick, we can finally obtain Equation (B.5).                               $\square$

 # C   BiPaRS for Deterministic Policy Setting

In this section, we define the BiPaRS problem for deterministic policy gradient algorithms. We provide a theorem similar to Theorem 1 and give the proof. Then we show the three methods explicit mapping (EM), meta-gradient learning (MGL), and incremental meta-gradient learning (IMGL) for approximating the gradient of the expected true reward with respect to the shaping weight function parameter $\phi$ in the deterministic policy setting.

Let $\mathcal{M} = \langle \mathcal{S}, \mathcal{A}, P, r, p_0, \gamma \rangle$ denote an MDP, $\mu_\theta$ denote an agent's deterministic policy with parameter $\theta$, $f$ denote the shaping reward function, and $z_\phi$ denote the shaping weight function with parameter $\phi$. Given $f$ and $z_\phi$, the agent should maximize the expected modified reward $\tilde{J}(\mu_\theta) = \mathbb{E}_{s \sim \rho^\mu} \big[ \big( r(s,a) + z_\phi(s,a) f(s,a) \big) \big|_{a=\mu_\theta(s)} \big]$. The objective of the shaping weight function $z_\phi$ is the expected accumulative true reward $J(z_\phi) = \mathbb{E}_{s \sim \rho^\mu} \big[ r(s,a) \big|_{a=\mu_\theta(s)} \big]$. Formally, the bi-level optimization of parameterized reward shaping (BiPaRS) problem for the deterministic policy setting can be defined as

$$
\begin{aligned}
\max_\phi \ & \mathbb{E}_{s \sim \rho^\mu} \big[ r(s,a) \big|_{a=\mu_\theta(s)} \big] \\
\text{s.t. } & \phi \in \Phi \\
& \theta = \arg\max_{\theta'} \ \mathbb{E}_{s \sim \rho^\mu} \big[ \big( r(s,a) + z_\phi(s,a) f(s,a) \big) \big|_{a=\mu_{\theta'}(s)} \big] \\
& \text{s.t. } \theta' \in \Theta.
\end{aligned}
\tag{C.6}
$$

The following theorem shows how to compute the gradient of the upper-level objective $J(z_\phi)$ with respect to the variable $\phi$ in Equation (C.6).

**Theorem 2.** *Given the shaping weight function $z_\phi$ and the deterministic policy $\mu_\theta$ of the agent in the upper level of the BiPaRS problem Equation (C.6), the gradient of the objective function $J(z_\phi)$ with respect to the shaping weight function parameter $\phi$ is*

$$
\nabla_\phi J(z_\phi) = \mathbb{E}_{s \sim \rho^\mu} \big[ \nabla_\phi \mu_\theta(s) \nabla_a Q^\mu(s,a) \big|_{a=\mu_\theta(s)} \big],
\tag{C.7}
$$

*where $Q^\mu$ is the state-action value function of $\mu_\theta$ in the original MDP.*

To prove the Theorem, we assume that $P(s,a,s')$, $\nabla_a P(s,a,s')$, $r(s,a)$, $\nabla_a r(s,a)$, $p_0(s)$ are continuous w.r.t. the variables $s$, $a$, and $s'$, and $\mu_\theta(s)$ are continuous w.r.t. $s$, $\theta$, and $\phi$. We also assume that the rewards are bounded. The proof is as follows.

*Proof.* Given a deterministic policy $\mu_\theta$, for any state $s$, the gradient of the state value $V^\mu(s)$ with respect to $\phi$ is

$$
\begin{aligned}
\nabla_\phi V^\mu(s) &= \nabla_\phi Q^\mu(s, \mu_\theta(s)) \\
&= \nabla_\phi \Big( r(s, \mu_\theta(s)) + \gamma \int_\mathcal{S} P(s, \mu_\theta(s), s') V^\mu(s') \mathrm{d}s' \Big) \\
&= \nabla_\phi \mu_\theta(s) \nabla_a r(s,a)|_{a=\mu_\theta(s)} + \nabla_\phi \Big( \gamma \int_\mathcal{S} P(s, \mu_\theta(s), s') V^\mu(s') \mathrm{d}s' \Big) \\
&= \nabla_\phi \mu_\theta(s) \nabla_a r(s,a)|_{a=\mu_\theta(s)} + \gamma \int_\mathcal{S} \nabla_\phi \mu_\theta(s) \nabla_a P(s,a,s')|_{a=\mu_\theta(s)} V^\mu(s') \mathrm{d}s' \\
&\quad + \gamma \int_\mathcal{S} P(s, \mu_\theta(s), s') \nabla_\phi V^\mu(s') \mathrm{d}s' \\
&= \nabla_\phi \mu_\theta(s) \nabla_a Q^\mu(s,a)|_{a=\mu_\theta(s)} + \int_\mathcal{S} \gamma p(s \to s', 1, \mu_\theta) \nabla_\phi V^\mu(s') \mathrm{d}s'.
\end{aligned}
$$

115 By expanding $\nabla_\phi V^\mu(s')$, we have

$$
\begin{aligned}
\nabla_\phi V^\mu(s) &= \nabla_\phi \mu_\theta(s) \nabla_a Q^\mu(s,a)|_{a=\mu_\theta(s)} + \int_\mathcal{S} \gamma p(s \to s', 1, \mu_\theta) \nabla_\phi Q^\mu(s', \mu_\theta(s')) \mathrm{d}s' \\
&= \nabla_\phi \mu_\theta(s) \nabla_a Q^\mu(s,a)|_{a=\mu_\theta(s)} + \int_\mathcal{S} \gamma p(s \to s', 1, \mu_\theta) \nabla_\phi \mu_\theta(s') \nabla_a Q^\mu(s', a)|_{a=\mu_\theta(s')} \mathrm{d}s' \\
&\quad + \int_\mathcal{S} \gamma p(s \to s', 1, \mu_\theta) \int_\mathcal{S} \gamma p(s' \to s'', 1, \mu_\theta) \nabla_\phi V^\mu(s'') \mathrm{d}s'' \mathrm{d}s' \\
&= \nabla_\phi \mu_\theta(s) \nabla_a Q^\mu(s,a)|_{a=\mu_\theta(s)} \\
&\quad + \int_\mathcal{S} \gamma p(s \to s', 1, \mu_\theta) \nabla_\phi \mu_\theta(s') \nabla_a Q^\mu(s', a)|_{a=\mu_\theta(s')} \mathrm{d}s' \\
&\quad + \int_\mathcal{S} \gamma^2 p(s \to s', 2, \mu_\theta) \nabla_\phi \mu_\theta(s') \nabla_a Q^\mu(s', a)|_{a=\mu_\theta(s')} \mathrm{d}s' \\
&\quad + \cdots \\
&= \int_\mathcal{S} \sum_{t=0}^\infty \gamma^t p(s \to s', t, \mu_\theta) \nabla_\phi \mu_\theta(s') \nabla_a Q^\mu(s', a)|_{a=\mu_\theta(s')} \mathrm{d}s'.
\end{aligned}
$$

116 Taking expectation over the initial states, we can obtain

$$
\begin{aligned}
\nabla_\phi J(z_\phi) &= \int_\mathcal{S} p_0(s) \nabla_\phi V^\mu(s) \mathrm{d}s \\
&= \int_\mathcal{S} p_0(s) \int_\mathcal{S} \sum_{t=0}^\infty \gamma^t p(s \to s', t, \mu_\theta) \nabla_\phi \mu_\theta(s') \nabla_a Q^\mu(s', a)|_{a=\mu_\theta(s')} \mathrm{d}s' \mathrm{d}s \\
&= \int_\mathcal{S} p_0(s) \int_\mathcal{S} \sum_{t=0}^\infty \gamma^t p(s \to s', t, \mu_\theta) \mathrm{d}s \nabla_\phi \mu_\theta(s') \nabla_a Q^\mu(s', a)|_{a=\mu_\theta(s')} \mathrm{d}s' \\
&= \int_\mathcal{S} \rho^\mu(s) \nabla_\phi \mu_\theta(s) \nabla_a Q^\mu(s, a)|_{a=\mu_\theta(s)} \mathrm{d}s,
\end{aligned}
$$

117 which is exactly Equation (C.7). □

## C.1 Explicit Mapping

119 The explicit mapping method makes the shaping weight function $z_\phi$ an input of the policy $\mu_\theta$.
120 Specifically, the policy $\mu_\theta$ is redefined as a hyper policy $\mu_\theta : \mathcal{S}_z \to \mathcal{A}$, where $\mathcal{S}_z = \{(s, z_\phi(s))|\forall s \in$
121 $\mathcal{S}\}$. According to the chain rule, we have

$$
\begin{aligned}
\nabla_\phi J(z_\phi) &= \mathbb{E}_{s \sim \rho^\mu} \left[ \nabla_\phi \mu_\theta(s, z) \nabla_a Q^\mu(s, a) \big|_{a=\mu_\theta(s,z), z=z_\phi(s)} \right] \\
&= \mathbb{E}_{s \sim \rho^\mu} \left[ \nabla_z \mu_\theta(s, z) \nabla_\phi z_\phi(s) \nabla_a Q^\mu(s, a) \big|_{a=\mu_\theta(s,z), z=z_\phi(s)} \right].
\end{aligned}
\tag{C.8}
$$

## C.2 Meta-Gradient Learning

123 Let $\theta$ and $\theta'$ be the policy parameters before and after one round of low-level optimization, respectively.
124 Let $\tilde{Q}$ denote the state-action value function under the policy $\mu_\theta$ in the modified MDP $\mathcal{M}' =$
125 $\langle \mathcal{S}, \mathcal{A}, P, \tilde{r}, p_0, \gamma \rangle$. We still assume that a batch of $N$ ($N > 0$) samples $\mathcal{B} = \{(s_i, a_i)|i = 1, ..., N\}$
126 is used to update $\theta$. According to the deterministic policy gradient theorem, we have

$$
\theta' = \theta + \alpha \sum_{i=1}^N \nabla_\theta \mu_\theta(s_i) \nabla_a \tilde{Q}(s_i, a)|_{a=\mu_\theta(s_i)},
\tag{C.9}
$$

where $\alpha$ is the learning rate. By taking the gradient of the both sides of Equation (C.9) with respect to $\phi$, we get

$$
\begin{aligned}
\nabla_\phi \theta' &= \nabla_\phi \Big( \theta + \alpha \sum_{i=1}^{N} \nabla_\theta \mu_\theta(s_i) \nabla_a \tilde{Q}(s_i, a)|_{a=\mu_\theta(s_i)} \Big) \\
&\approx \alpha \sum_{i=1}^{N} \nabla_\theta \mu_\theta(s_i)^\top \nabla_\phi \big( \nabla_a \tilde{Q}(s_i, a)|_{a=\mu_\theta(s_i)} \big),
\end{aligned}
\tag{C.10}
$$

where $\theta$ is treated as a constant with respect to $\phi$. However, for each sample $i$ in the batch $\mathcal{B}$, we cannot directly compute the value of the term $\nabla_\phi \big( \nabla_a \tilde{Q}(s_i, a)|_{a=\mu_\theta(s_i)} \big)$ even we replace $\tilde{Q}(s_i, a)$ by a Monte Carlo return as in the stochastic policy case. We adopt the idea of the explicit mapping method to solve this problem. That is, to include $z_\phi(s)$ as an input of $\tilde{Q}$. As we discuss in Section 4.1, this makes $\tilde{Q}$ the state-action value function of $\mu_\theta$ in an equivalent MDP $\tilde{\mathcal{M}}_z = \langle \mathcal{S}_z, \mathcal{A}, P_z, \tilde{r}_z, p_z, \gamma \rangle$ and is transparent to the agent. For simplicity, we denote $\delta(s_i, a, z) = \nabla_a \tilde{Q}(s_i, a, z)$. With the extended state-action value function, we have

$$
\begin{aligned}
\nabla_\phi \theta' &\approx \alpha \sum_{i=1}^{N} \nabla_\theta \mu_\theta(s_i)^\top \nabla_\phi \big( \nabla_a \tilde{Q}(s_i, a, z)|_{a=\mu_\theta(s_i), z=z_\phi(s_i)} \big) \\
&= \alpha \sum_{i=1}^{N} \nabla_\theta \mu_\theta(s_i)^\top \nabla_\phi \delta(s_i, a, z)|_{a=\mu_\theta(s_i), z=z_\phi(s_i)} \\
&= \alpha \sum_{i=1}^{N} \nabla_\theta \mu_\theta(s_i)^\top \nabla_z \delta(s_i, a, z)|_{a=\mu_\theta(s_i), z=z_\phi(s_i)} \nabla_\phi z_\phi(s_i).
\end{aligned}
\tag{C.11}
$$

### C.3 Incremental Meta-Gradient Learning

In Equation (C.10), we can also treat $\theta$ as a non-constant with respect to $\phi$ because $\phi$ is optimized according to the old policy $\pi_\theta$ in the last round of upper-level optimization. For the simplification purpose, we let $g_\theta(s) = \nabla_\theta \mu_\theta(s)$. Then Equation (C.10) can be rewritten as

$$
\begin{aligned}
\nabla_\phi \theta' &= \nabla_\phi \Big( \theta + \alpha \sum_{i=1}^{N} \nabla_\theta \mu_\theta(s_i) \nabla_a \tilde{Q}(s_i, a)|_{a=\mu_\theta(s_i)} \Big) \\
&= \nabla_\phi \theta + \alpha \sum_{i=1}^{N} \Big( \nabla_\phi g_\theta(s_i)^\top \nabla_a \tilde{Q}(s_i, a) + g_\theta(s_i)^\top \nabla_\phi \big( \nabla_a \tilde{Q}(s_i, a) \big) \Big) \Big|_{a=\mu_\theta(s_i)}
\end{aligned}
\tag{C.12}
$$

For computing the value of $\nabla_\phi \big( \nabla_a \tilde{Q}(s_i, a)|_{a=\mu_\theta(s_i)} \big)$, once again we can include $z_\phi$ in the input of $\tilde{Q}$. Denoting $\nabla_a \tilde{Q}(s_i, a, z)$ by $\delta(s_i, a, z)$, we have

$$
\nabla_\phi \theta' \approx \nabla_\phi \theta + \alpha \sum_{i=1}^{N} \Big( \nabla_\theta g_\theta(s_i)^\top \nabla_\phi \theta\, \delta(s_i, a, z) + g_\theta(s_i)^\top \nabla_\phi \delta(s_i, a, z) \Big) \Big|_{a=\mu_\theta(s_i), z=z_\phi(s_i)}
\tag{C.13}
$$

By substituting Equation (C.13) into Equation (C.7), we can get the third approximation of the gradient $\nabla_\phi J(z_\phi)$. In fact, in Equation (C.12) we can treat the shaping state-action value $\tilde{Q}$ as a constant with respect to $\phi$ so that we do not have to extend the input space of $\tilde{Q}$ and can get another version of $\nabla_\phi \theta'$, namely

$$
\nabla_\phi \theta' \approx \nabla_\phi \theta + \alpha \sum_{i=1}^{N} \nabla_\theta g_\theta(s_i)^\top \nabla_\phi \theta\, \nabla_a \tilde{Q}(s_i, a)|_{a=\mu_\theta(s_i)}.
\tag{C.14}
$$

Furthermore, we can also remove the second-order term $\alpha \sum_{i=1}^{N} \nabla_\theta g_\theta(s_i)^\top \nabla_\phi \theta\, \delta(s_i, a, z)$ of Equation (C.13) and get a computationally cheaper approximation

$$
\nabla_\phi \theta' \approx \nabla_\phi \theta + \alpha \sum_{i=1}^{N} \Big( g_\theta(s_i)^\top \nabla_z \delta(s_i, a, z) \nabla_\phi z_\phi(s_i) \Big) \Big|_{a=\mu_\theta(s_i), z=z_\phi(s_i)}.
\tag{C.15}
$$

## D  Experiments

In this section, we provide the details of the problem and algorithm hyperparameter settings of the *cartpole* and *MuJoCo* experiments.

### D.1  Cartpole

**Problem Setting:** We choose the *cartpole* task from the OpenAI Gym-v1 benchmark. The cartpole system consists of a pole and a cart. The pole is connected to the cart by an un-actuated joint and the cart can be controlled by the agent to move along the horizontal axis. Each episode starts by setting the position of the cart randomly within the interval $[-0.05, 0.05]$ and setting the angle between the pole and the vertical direction smaller than 3 degrees. In each step of an episode, the agent should apply a positive or negative force to the cart to let the pole remain within 12 degrees from the vertical direction and keep the position of the cart within $[-2.4, 2.4]$. An episode will be terminated if either of the two conditions is broken or the episode has lasted for 200 steps. In the discrete-action cartpole, the action space of the agent has only two actions $+1$ and $-1$, while in the continuous-action cartpole, the agent has to decide the specific value of the force to be applied.

**Hyperparameter Settings:** In the *cartpole* experiment, the base learner PPO adopts a two-layer policy network with 8 units in each layer and a two-layer value function network with 32 units in each hidden layer. Both the policy and value function networks adopt *relu* as the activation function in each hidden layer. The policy and value function networks are updated every $20,000$ steps and one such update contains 50 optimizing epochs with batch size 1024. The threshold for clipping the probability ratio is 0.5. We adopt the generalized advantage estimator (GAE) to replace Q-function for computing policy gradient and the hyperparameter $\lambda$ for bias-variance trade-off is 0.95.

The DPBA method adopts a neural network to learn potentials from shaping rewards, which has two full-connected (FC) *tanh* hidden layers with 16 units in the first layer and 8 units in the second layer. The BiPaRS methods also use two-layer neural network to represent the shaping weight function, with 16 and 8 units in the first and second layers and *tanh* as the activation in both layers. The outputs of the shaping weight function network for all state-action pairs are initialized to 1 using the following way. Firstly, the weights and biases of the hidden layers are initialized from a uniform distribution in $[-0.125, 0.125]$. Secondly, the weights and bias of the output layer are initialized randomly uniformly in $[-10^{-3}, 10^{-3}]$. The two steps make sure that the outputs are near zero. Finally, we add 1 to the output layer so that the initial shaping weights of all state-action pairs are near 1.

All these networks are optimized using the Adam optimizer. The learning rates for optimizing the policy and the value function networks are $10^{-4}$ and $2 \times 10^{-4}$, respectively. The learning rate for updating the potential network of the DPBA method is $5 \times 10^{-4}$. The learning rate for optimizing the shaping weight function of the BiPaRS methods is set differently in different tests. In the tests with the original shaping reward function, the learning rate is $10^{-5}$, while in the tests with the harmful shaping reward function (i.e., the first adaptability test) and the random shaping reward function (i.e., the third adaptability test), it is $5 \times 10^{-4}$. Recall that both the harmful and random shaping reward functions make it difficult to learn a good policy. We set the learning rate higher in the two tests to make the shaping weights change rapidly from the initial value 1.0. The discount rate $\gamma$ is 0.999 in all tests. We summarize the above hyperparameter settings in Table 1. Note that the naive shaping (NS) method directly adds the shaping reward to the original reward function, so it has no other hyperparameters.

### D.2  MuJoCo

Recall that in the *MuJoCo* experiment, we adopt the shaping reward function $f(s, a) = w(0.25 - \frac{1}{L}\sum_{i=1}^{L}|a_i|)$ to limit the average torque amount of the agent's action, where $w$ is a task-specific weight which makes $f(s, a)$ have the same scale as the true reward in the initial learning phase. Now we show the value of $w$ in each task in Table 2.

Since the MuJoCo tasks are more complicated than the cartpole task, the policy and value function networks of the base learner PPO have three hidden layers with 64 units and *relu* activation in each layer. The RCPO algorithm also adopts PPO as the base learner and uses the same network architectures. The potential network of DPBA has two *tanh* hidden layers and each layer also has 64

| Algorithm | Hyperparameters |
|---|---|
| Base Learner (PPO) | policy network: two 8-unit FC hidden layers, *relu* activation<br>value network: two 32-unit FC hidden layers, *relu* activation<br>threshold of probability ratio clipping ($\epsilon$): 0.5<br>update period: $20,000$ steps<br>number of epoches per update: 50<br>batch size: 1024<br>GAE parameter ($\lambda$): 0.95<br>optimizer: Adam<br>learning rate of policy network: $10^{-4}$<br>learning rate of value network: $2 \times 10^{-4}$<br>discount rate ($\gamma$): 0.999 |
| DPBA | potential network: 16-unit FC layer + 8-unit FC layer, *tanh* activation<br>optimizer: Adam<br>learning rate of potential network: $5 \times 10^{-4}$ |
| BiPaRS | shaping weight network: 16-unit FC layer + 8-unit FC layer, *tanh* activation<br>optimizer: Adam<br>learning rate of shaping weight network: $10^{-5}$ in the original test, and $5 \times 10^{-4}$ in the adaptability tests<br>initial shaping weight: 1.0<br>shaping weight range: $(-\infty, +\infty)$ |

Table 1: The hyperparameters of the tested algorithms in the *cartpole* experiment

units. The shaping weight function network of our BiPaRS methods has two *tanh* hidden layers, with 16 and 8 units in the first and second layers, respectively.

All the tested algorithms perform model update every $20,000$ training steps. One updating process contains 50 epochs and each updating epoch uses a batch of 1024 samples. The threshold for clipping the probability ratio is 0.2 in all of the five MuJoCo tasks. The discount rate $\gamma$ is 0.999 and the parameter $\lambda$ of GAE is 0.95. The Lagrange multiplier of RCPO is bounded in $[0, 10000]$. The neural network models of all algorithms are optimized by the Adam optimizer. The learning rates for optimizing the policy and value function networks are $10^{-4}$ and $2 \times 10^{-4}$, respectively. The learning rates for updating the potential network of DPBA and the shaping weight function of BiPaRS-EM are $5 \times 10^{-4}$. The BiPaRS-MGL and BiPaRS-IMGL algorithms use a learning rate of $10^{-3}$ to update their shaping weight functions. The RCPO algorithm uses a dynamic learning rate to update its Lagrange multiplier, which starts from $5 \times 10^{-5}$ and exponentially decays with a factor $1 - 10^{-9}$.

One thing should be noted is that the *Humanoid-v2* task has a much higher state dimension than the other four tasks, which means that the neural networks of the tested algorithms have much more parameters. Therefore, for training the BiPaRS-IMGL algorithm in *Humanoid*, directly approximating the meta-gradient $\nabla_\phi \theta'$ according to Equation (A.4) is extremely difficult because the Hessian matrix $\nabla_\theta g_\theta(s_i, a_i)^\top$ with respect to the policy parameter $\theta$ has $\mathcal{O}(n^3)$ computational complexity. To address this issue, we make the policy network a two-layer network with only 32 units in each layer and ignore the term $\alpha \sum_{i=1}^N \tilde{Q}(s_i, a_i) \nabla_\theta g_\theta(s_i, a_i)^\top$ in Equation (A.4). We summarize all the hyperparameter settings in the *MuJoCo* experiment in Table 3.

| | Swimmer-v2 | Hopper-v2 | Humanoid-v2 | Walker2d-v2 | HalfCheetah-v2 |
|---|---|---|---|---|---|
| $w$ | 20 | 16 | 20 | 10 | 100 |

Table 2: The task-specific weight $w$ in each of five *MuJoCo* tasks

Assume that there are $M$ monks and the total amount of the rice gruel is $L$. Without loss of generality, we assume that the rice gruel is assigned to the monks according to the order $1, 2, ..., M$. The decision-making at each step is two-fold. Firstly, the system chooses one from the $M$ monks. Secondly, the chosen monk is the assigner, who should determine the amount of rice gruel to the current monk to be assigned.

Formally, this can be modeled as follows. At each step $t$ ($t = 1, 2, ..., M$), the system first chooses a monk $m_t \in \{1, ..., M\}$. Then the chosen monk has to choose an action $l_t \in [0, L - \sum_{t'=1}^{t-1} l_{t-1}]$, which stands for the amount of rice gruel assigned to the monk at the assign order $t$. Therefore, the problem has only $M + 1$ states and the state transitions are deterministic:

$$S_1 \rightarrow S_2 \rightarrow \cdots \rightarrow S_M \rightarrow S_T, \tag{C.16}$$

where $S_T$ is the terminal state.

There are $M + 1$ agents in the system, namely the $M$ monks and the system. Importantly, we assume that each monk tries to maximize its own utility, while the sysytem tries to make the utility of each monk the same. For any step $t$, let $s_t$ denote the state and $a_t = (m_t, l_t)$ denote the joint action of the system and the chosen monk. Then for any monk $i \in \{1, ..., M\}$, the reward function is

$$R_i(s_t, a_t) = R_i(s_t, m_t, l_t) = \begin{cases} l_t & \text{if } i = t, \\ 0 & \text{o.w.}, \end{cases} \tag{C.17}$$

which means that only the monk at the assign order $t$ gets the rice gruel assigned by monk $m_t$. The system reward function is

$$R_0(s_t, a_t) = R_0(s_t, m_t, l_t) = -|\frac{L}{M} - l_t|, \tag{C.18}$$

which stands for the absolute fairness of the system.

| Algorithm | Hyperparameters |
|---|---|
| Base Learner (PPO) | policy network: three 64-unit FC hidden layers, *relu* activation<br>value network: three 64-unit FC hidden layers, *relu* activation<br>threshold of probability ratio clipping ($\epsilon$): 0.2<br>update period: $20,000$ steps<br>number of epoches per update: 50<br>batch size: 1024<br>GAE parameter ($\lambda$): 0.95<br>optimizer: Adam<br>learning rate of policy network: $10^{-4}$<br>learning rate of value network: $2 \times 10^{-4}$<br>policy gradient clip norm: 1.0<br>value fuction gradient clip norm: 1.0<br>discount rate ($\gamma$): 0.999 |
| DPBA | potential network: two 64-unit FC layers, *tanh* activation<br>optimizer: Adam<br>learning rate of potential network: $5 \times 10^{-4}$<br>potential network gradient clip norm: 10.0 |
| BiPaRS | shaping weight network: 16-unit FC layer + 8-unit FC layer, *tanh* activation<br>optimizer: Adam<br>learning rate of shaping weight network: $5 \times 10^{-4}$ for BiPaRS-EM, and $5 \times 10^{-4}$ for BiPaRS-MGL and BiPaRS-IMGL<br>shaping weight network gradient clip norm: 10.0<br>initial shaping weight: 1.0<br>shaping weight range: $[-1, 1]$ |
| RCPO | Lagrange multiplier lower bound: 0<br>Lagrange multiplier upper bound: 10000<br>initial learning rate of Lagrange multiplier: $5 \times 10^{-5}$<br>decay factor of learning rate: $1 - 10^{-9}$ |
| Other | policy network of BiPaRS-IMGL in *Humanoid-v2*: two 32-unit FC hidden layers, *relu* activation |

Table 3: The hyperparameters of the tested algorithms in the *MuJoCo* experiment