[Reviews · NeurIPS 2020]

Review 1

Summary and Contributions: The authors focus on cases where a hand-crafted shaping reward is available, but the agent designer is unsure of whether and where it is useful, and so one would like to learn a state-action-dependent weighting of the shaped reward. The authors present three methods of estimating the gradients for updating those weights. Experimentally, they evaluate their method on Cartpole and in MuJoCo. For Cartpole, they test whether their methods can learn to upweight useful shaping rewards and downweight harmful or random rewards. Additionally, they test the transfer of learnt reward shaping weights to another agent trained from scratch. In MuJuCo, they only test whether an agent can learn to ignore harmful shaping rewards.

Strengths: The introduction provided clear motivation for the problem they were solving. The background and related work were clear and answered questions I had just as I asked them. The introduction of their method in sections 3 and 4 was very clear. I found the problem they were solving interesting and important, and their solution sound and well-explained.

Weaknesses: Given how strong the first four sections (five pages) of the paper were, I was relatively disappointed in the experiments, which were somewhat light. Specifically: 1) While the authors' methods allow for learning a state-action-dependent weighting of the shaping rewards, it seemed to me possible that in all of the experiments presented, learning a *uniform* state-action-independent weighting would have sufficed. Moreover, since learning a state-action-independent weighting is much simpler (i.e. it is a single scalar), it may even outperform the authors' methods for the current experiments. Based on this, I would like to suggest the following: 1a) Could the authors provide visualizations of the state-action variation of their learnt weightings? They plot the average weight in some cases (Fig 1 and 3), but given Cartpole has such a small state-action space, it should be possible to visualize the variation. The specific question here is: do the weights vary much at all in these cases? 1b) Could the authors include a baseline of learnt state-action-*independent* weights? In other words, this model has a single parameter, replacing z_phi(s,a) with a single scalar z. This should be pretty easy to implement. The authors could take any (or all) of their existing gradient approximators and simply average them across all (s,a) in a batch to get the gradient w.r.t. z. 1c) Could the authors include an additional experiment that specifically benefits from learning state-action-*dependent* (so non-uniform) weights? Here is a simple example for Cartpole: the shaping reward f(s,a) is helpful for half the state space and unhelpful for the other half. The "halves" could be whether the pole orientation is in the left or right half. The helpful reward could be that from Section 5.1 while the unhelpful reward could be that from the first adaptability test in Section 5.3. 2) To me, the true power of the author's approach is not in learning to ignore bad rewards (just turn them off!) but to intelligently incorporate sort-of-useful-but-not-perfect rewards. This way a researcher can quickly hand design an ok shaping reward but then let the authors' method transform it into a good one. Thus, I was surprised the experiments focussed primarily on ignoring obviously bad rewards and upweighting obviously good rewards. In particular, the MuJoCo experiments would be more compelling if they included more than just a single unhelpful shaping reward. I think the authors could really demonstrate the usefulness of their method there by doing the following: hand design a roughly ok shaping reward for each task. For example, the torso velocity or head height off the ground for Humanoid-v2. Then apply the authors' method and show that it outperforms naive use of this shaping reward. 3) Although the authors discussed learning a shaping reward *from scratch* in the related work section, I was surprised that they did not included this as a baseline. One would like to see that their method, when provided with a decent shaping reward to start, can learn faster by leveraging this hand-crafted knowledge. Fortunately, it seems to me again very easy to implement a baseline like this within the author's framework: simply set f(s,a)=1 and use the authors' methods (perhaps also initializing z_phi(s,a)=0).

Correctness: The derivation of the method looked correct to me and was well-explained. Some small typos: 1) Equation 8: shouldn't t -> t+j? Otherwise the only thing changing in the terms of the sum is the discount factor. Should also probably add time indices to the LHS of the equation as well as g_theta for clarity. 2) Footnote 1 on page 3: between -> from 3) Line 232: change to "benefit the maximization of the true rewards."

Clarity: Generally speaking, the paper is very clearly written. Some small comments: 1) The plots would be easier to read if they all used the same legend. Currently, the legends switches even within a figure, which is very confusing. I recommend using a styling such that the authors' methods are easily grouped together and distinguished from the baselines, such as in Fig 1a left where the authors' methods are denoted by solid lines and the baselines by dashed lines. 2) Why are these only error bars in Figure 2? Are the error bars in Fig 2 across seeds? Did the authors not run multiple seeds for Cartpole? There should be error bars for all of the experiments. 3) Section 5.2: There are exactly 9 words devoted to RCPO. What is RCPO? Why is it introduced as a baseline? Why is it only used for MuJoCo? 4) I'm not a fan of the BiPaRS acronym. Hard to pronounce and looks a bit silly to me. 5) Section 5.3: I think its slightly misleading to call the reward in the first test "harmful." There is useful signal there; the shaping reward just needs inverted. This is sort of mentioned later in lines 271-272 but it should be clarified at the beginning. 6) Fig 1 and 3: The x-axes should be consistent. Moreover, its strange to use a scale of 1.2e6. Why not just use 1e6? 7) Fig 4 captions: "The results of the second/third adaptability test" are not very helpful captions. I would use "Transfer of reward weights test" and "Random rewards test" or something like that. 8) Lines 264-265: Should remind the reader that PPO from experiment 5.1 is trained with useful shaping rewards and also explain why it is the appropriate comparison.

Relation to Prior Work: Mostly the paper was excellent in this regard. However, two papers I would like to see discussed: 1) The DeepMind "Capture The Flag" paper: https://science.sciencemag.org/content/364/6443/859.full?ijkey=rZC5DWj2KbwNk&keytype=ref&siteid=sci. This paper is relevant because it seems to me one of the bigger successes in learning shaping rewards from scratch. 2) The UC-Berkeley "Should Robots Be Obedient?" paper: https://people.eecs.berkeley.edu/~dhm/papers/IJCAI17_obedience.pdf. This paper is relevant because one interpretation of the agent learning how to weight hand-crafted shaping rewards is the agent determining to what extent *to follow a human command.* This could also suggest other application domains for the authors' methods.

Reproducibility: Yes

Additional Feedback: Update after author rebuttal: Thank you to the authors for the additional experiments. Figures 1a-c are useful in addressing my points on state-action (in)dependence. I may be misunderstanding, but I don't think 1d is what I had in mind about "learning from scratch." By learning from scratch, I meant setting the shaping reward to a constant value of 1, and then learning the weights from zero initialization, not keeping the informative shaping reward from learning zero init. In this way, all of the "shaping" would be in the learnt weights. I have raised my score from a 6 to a 7, based on the additional experiments, with the assumption that the authors will clarify 1d and if necessary rerun the baseline.


Review 2

Summary and Contributions: Reward shaping is a way of using domain knowledge to speed up convergence of reinforcement learning algorithms. Shaping rewards designed by domain experts are not always accurate, and they can hurt performance or at least provide only limited improvement. In this paper, the authors assume that the shaping function, f(s,a), is given by the domain expert, and the goal of the methods proposed in this paper is to learn how to maximally exploit the function, f(s,a). To achieve this goal, the authors consider direct policy search methods, and they propose a two-level optimisation problem. Direct policy optimisation methods use gradient descent, and the authors had to propose a way of computing gradients for their two-level model. Empirical results are presented along with a detailed study of what happens when the shaping function is highly incorrect.

Strengths: - The problem addressed in this paper is important and relevant to NeurIPS. - The methods are algorithmically interesting and non-trivial. - Empirical results are appropriate; many factors were considered in evaluation, e.g., the quality of the shaping function and the robustness of the methods proposed in this paper.

Weaknesses: The DPBA method used in comparisons is a potential-based method (at least potential-based is in its name), whereas the methods proposed in this paper are not in this family. Perhaps the authors could clarify this fact in their results section, i.e., please explain which methods are potential-based and which are not. "bi-level optimisation" is a nice name invented by the authors, but it seems to me that this approach is related to standard methods in optimisation, e.g., this is probably an instance of coordinate descent (https://en.wikipedia.org/wiki/Coordinate_descent). I believe that the authors should link their approach with the relevant approaches in optimisation. Without seeing further explanations in this paper, I believe that this approach is just an example of a standard coordinate descent.

Correctness: The supplementary material is long, and I did not read it, but the main paper looks correct.

Clarity: The paper is very well written. It is definitely in the top half of the papers that I am reviewing for this conference. The authors managed to compress a lot of technical material into 8 pages, yet the paper reads very well.

Relation to Prior Work: Prior work is sufficient. I was initially concerned that [18] and [25] solved this problem, but the authors have a good explanation in line 96. Relevant papers on potential-based reward shaping are cited, but the authors could still clarify that their methods are not potential based. This is not a problem at all for me because not all reward shaping methods have to be potential-based, but this clarification would be useful for the future readers.

Reproducibility: No

Additional Feedback: After reading your response, I still think that this is a very good paper. I was not sure however how to argue with the other reviews about this statement in your paper: "In the function approximation setting, how to utilize the shaping rewards to learn a good policy seems more important than just keeping policy invariant." This property of your method is indeed problematic.


Review 3

Summary and Contributions: Shaped reward can make learning easier and safer, but sometimes it can also produce suboptimal policies. This paper asks the key question: can we automatically make sure that the shape reward gives the same policy as the original one?

Strengths: The analysis is fairly thorough and complete. The argument that you can meta-learn a weight to recover from illy-specified reward shaping is well-supported. I particular like the results on the learning dynamics of a meta-learned shaping weight.

Weaknesses: I like the general idea and the presentation, but there are a few things that bugs me: 1. methodologically, they show the performance of the learned agent. Reward in [21] sometimes gives sub-optimal policy because the constrained policy is actually desired for specific concerns. It is not clear if getting higher reward in violation of the torque constrain is in fact desired. In some way, it looks like learning a correction of the shaped reward that incorporates safety constraints just gets us back to square one. It is understandable that some form of automatic reward shaping *would be helpful*. It is just that the experiment here are a little bit contrived. So my question is: "if this allows one to recover the reward before shaping, how does this affect safety that was the point of the original reward constrained policy optimization?" 2. The three methods are not really well distinguished empirically, and there lack discussion and theoretical insight on why and when one would be preferred over another. I am not sure if I gained much insight from the results regarding the three variants. I do have questions about the variance of the IMGL variant, but the variance between the MuJoCo environments makes it hard to draw conclusions. 3. The formulation of policy gradient does not involve a learned value baseline. This could be okay for the domains used in this paper, but how about the some of the harder tasks where a baseline is needed? Would this affect/limit the applicability of the meta-gradient?

Correctness: Yes, but I wish they provide more details on the reward shaping term in the reward constrained PO as part of the paper, and provide more theoretical support for why one version of their algorithm would be better than another.

Clarity: The paper is fairly well written, but I think it could be better motivated, and the argument why this is necessary, and the trade off against the safety benefit of a constrained reward shaping term can be made more explicit.

Relation to Prior Work: yes, but see above.

Reproducibility: Yes

Additional Feedback: - Fig.1: the color scheme is hard to look at. I would change it to something that is less saturated. Font size is in generally too small. - Is the Shaping Weight in log scale in Fig.3? I am wondering about this because it goes down to -100 which is quite large.


Review 4

Summary and Contributions: This paper proposes to learn to weight the shaping rewards to optimize for the original reward function. Its contributions are the formulation as bi-level optimization and three proposed ways to compute gradients in the upper level.

Strengths: The idea is natural and makes sense. The paper is well written with code.

Weaknesses: Though well written and could follow, the paper seems work still in progress with three algorithms proposed to compute the upper level gradient (somewhat ad-hoc, e.g. explicit mapping). More effort is expected to make the paper and its conclusions/takeaways more concentrated. The comparison with policy-invariant reward shaping is almost purely empirical: the PBRS family is guaranteed to be policy-invariant, while here it's only indirectly realized through the upper level optimization with only asymptotic guarantee under good enough learning. There should be more discussion/analysis w.r.t. PBRS family and other reward shaping methods, since for now many questions remain unclear. There's also a problem with the DPBA baseline: though published, it has been simply proven to be wrong - see [Behboudian et al. Useful Policy Invariant Shaping from Arbitrary Advice, AAMAS 2020 ALA workshop]. For the purpose of this paper's experiments, it seems sufficient to use the non-dynamic version (e.g. the original PBRS, or to include action [Wiewiora et al. Principled Methods for Advising Reinforcement Learning Agents, ICML 2003]). Therefore there's still much to be done in comparison with existing reward shaping works, both theoretically and empirically.

Correctness: Generally yes.

Clarity: Yes.

Relation to Prior Work: Not enough. Besides the points mentioned in "Weaknesses" regarding existing reward shaping, this paper also seems a special case of the optimal reward framework (only you're provided with a shaping function, but overall the upper level is still learning a reward function for the lower level)?

Reproducibility: Yes

Additional Feedback: ========== post-rebuttal ========== I've read the other reviews and the authors rebuttal. I'm still on the negative side on this paper. While it provides some promising empirical results, my main concerns (policy invariance and DPBA family-related questions, as in my original review) are not addressed by the rebuttal. I know the ideal non-parametric solution of the proposed BiPaRS is the original optimal policy (I've said before "here it's only indirectly realized through the upper level optimization with only asymptotic guarantee under good enough learning"). DRL is a difficult optimization problem in itself already. I don't think it's a good idea to make it more difficult (bi-level) unless there's more back-up. I cannot totally agree on the authors in rebuttal on "In the function approximation setting, how to utilize the shaping rewards to learn a good policy seems more important than just keeping policy invariant." It's rather debatable. The original reward is almost the most important thing that algorithm designers care about and of course they'll try to make it as close to their true goals as possible. So policy invariance is still a much desired property under reward shaping, and I would assume a "good" policy that's not policy invariant under RS has more to thank luck. Also, the authors haven't addressed the problem of the DPBA baseline at all. As I've mentioned, though published, it has been simply proven to be wrong. This makes the empirical results not ready enough for publication yet. Besides, as I've mentioned in "Weakness", I dislike that three algorithms are proposed to compute the upper level gradient, which makes the paper's takeaways less concentrated. I'll maintain my score as my original concerns haven't been addressed yet both theoretically and empirically.

[Author Response · NeurIPS 2020]



Figure 1: Results of the additional experiments. Figs 1(a) and 1(b) are the results of the single-weight baseline method in continuous carpole with beneficial and harmful shaping rewards, respectively. Fig 1(c) is the result of the second experiment and Fig 1(d) is the heat map of BiPaRS-EM's shaping weights in different states. Fig 1(e) shows the comparison between our methods with normal and zero initialization of the shaping weights

We thank all reviewers for the valuable comments and suggestions. Our responses to the main concerns are as follows.

1. **Limitation of the experiments** (proposed by reviewer 1): We conduct three additional experiments in the continuous
cartpole task to answer the reviewer's questions.

(1) In the cartpole experiment in Section 5.1 and the first adaptability test in Section 5.3, the shaping weights learnt by
our methods differ slightly across the state-action space. So learning a state-action-independent shaping weight in the
two tests is truly sufficient. We implement the baseline method which replaces the shaping weight function $z_\phi$ with a
single shaping weight and test it in the two experiments. The results are given in Figs 1(a) and 1(b) and it can be found
that the single-weight method outperforms the other methods.

(2) As suggested by reviewer 1, we also conduct an experiment where half of the shaping rewards are helpful and the
other half of the shaping rewards are harmful. It can be found from the results in Figure 1(c) that our methods perform
the best and the single-weight baseline method cannot perform as well as in Figures 1(a) and 1(b). We plot the shaping
weights learnt by the BiPaRS-EM method across a subset of the state space (containing 100 states) as a heat map in
Figure 1(d) to show that our methods are able to learn state-dependent or state-action-dependent shaping weights.

(3) We conduct the third experiment to test the baselines methods which learn shaping rewards from scratch. According
to reviewer 1's suggestion again, such baseline methods are simply implemented by our BiPaRS methods with zero
initialization of the shaping weights. For convenience, the setting of the shaping rewards is the same as the second
experiment and the results are shown in Figure 1(e). It can be found that when learning from scratch, all our methods
fail to learn as well as their normal versions where the shaping weights are initialized to 1. The zero initialization of the
shaping weights means that the prior knowledge incorporated in the shaping rewards is invisible to the algorithm in the
beginning and this may lead to more effort of exploration.

2. **Saftey concern in MuJoCo** (proposed by reviewer 3): In fact, we adopt the MuJoCo setting of the RCPO paper
in our experiment because it provides a good shaping reward function for the MuJoCo tasks. Although such shaping
reward function is originally used as constraints, we only care about whether our methods can obtain higher true rewards
and how they will do if the shaping rewards are in conflict with the true rewards.

3. **Comparison with PBRS method** (proposed by reviewer 4): The PBRS family mainly focuses on the guarantee of
policy invariant, and our methods are proposed for solving the utilization problem of given shaping rewards. Although
most PBRS methods have the policy invariant property, whether an optimal policy can be learnt by a learning algorithm
(especially a DRL algorithm) does not totally depend on this. In the function approximation setting, how to utilize the
shaping rewards to learn a good policy seems more important than just keeping policy invariant. Furthermore, BiPaRS
also has policy invariant property because the solution of its objective is an optimal policy of the original MDP.

4. **Relation to the optimal reward framework** (proposed by reviewer 4): Our methods are essentially different from
the optimal reward framework (ORF). Firstly, an ORF method such as the LIRPG algorithm (reference [25] in our
paper) is similar to our BiPaRS-MGL method. But our first method BiPaRS-EM, and the third method BiPaRS-IMGL,
cannot be directly derived from LIRPG. Secondly, all our methods are based on Theorem 1, which actually is more
general than Eq. (5) in the LIRPG paper. It may be better to say that "our methods and ORF are special cases of
meta-policy gradient methods" than to say that "our methods are special cases of ORF". The shaping weights learnt by
our methods are evaluation and guide of the utilization of the shaping rewards. Perhaps we can treat $z_\phi(s, a) f(s, a)$ as
an intrinsic reward, but $z_\phi(s, a)$ itself is not.

5. We will correct the typos and modify our paper according to the comments of the reviewers (e.g., improving the
related work section, providing comprehensive comparison between our methods, and adding more experiments).

[Meta-Review · NeurIPS 2020]

This paper proposes a method to learn shaping rewards in RL to improve learning. The authors clearly explain the problem and their method. The experimental results show clearly their method working as intended. I would expect the authors to update the final draft of their manuscript with the additional experiments provided in the author response and referencing and discussing the relation of their method to crucial pieces of prior work suggested by reviewers, in particular "Human-level performance in 3D multiplayer games with population-based reinforcement learning" which also performs bi-level optimisation of shaping rewards.